# Concentration and excess risk bounds for imbalanced classification with synthetic oversampling

**Touqeer Ahmad[1], Mohammadreza M. Kalan[1], François Portier[1], Gilles Stupfler[2]**

[1]Univ Rennes, Ensai, CNRS, CREST—UMR 9194, F-35000 Rennes, France
[2]Univ Angers, CNRS, LAREMA, SFR MATHSTIC, F-49000 Angers, France

## Abstract

Synthetic oversampling of minority examples using SMOTE and its variants is a leading strategy for addressing imbalanced classification problems. Despite the success of this approach in practice, its theoretical foundations remain underexplored. We develop a theoretical framework to analyze the behavior of SMOTE and related methods when classifiers are trained on synthetic data. We first derive a uniform concentration bound on the discrepancy between the empirical risk over synthetic minority samples and the population risk on the true minority distribution. We then provide a nonparametric excess risk guarantee for kernel-based classifiers trained using such synthetic data. These results lead to practical guidelines for better parameter tuning of both SMOTE and the downstream learning algorithm. Numerical experiments are provided to illustrate and support the theoretical findings.

## 1 Introduction

The problem of imbalanced classification arises when one of the classes of interest, often referred to as the *minority class*, is significantly underrepresented relative to the other classes. In such settings, standard classification algorithms tend to produce trivial decision rules that favor the majority class, often ignoring the minority class altogether. Designing algorithms that perform well under class imbalance has become a key challenge in modern machine learning [Chawla et al., 2004, He and Garcia, 2009, Lemaitre et al., 2017, Spelmen and Porkodi, 2018, Feng et al., 2021].

There are two main families of methods to tackle this problem. *Model-level approaches* modify the training objective, often by reweighting the loss function to penalize misclassification of the minority class more heavily. Several theoretical developments support this direction, including work on cost-sensitive learning [Menon et al., 2013, Xu et al., 2020, Aghbalou et al., 2024], Neyman-Pearson classification [Rigollet and Tong, 2011, Tong, 2013, Kalan and Kpotufe, 2024] and applications to deep learning models [Cao et al., 2019, Byrd and Lipton, 2019].

By contrast, *data-level approaches* modify the training data by applying resampling techniques [Kubat and Matwin, 1997, Chawla et al., 2002, Mani and Zhang, 2003, Barandela et al., 2004, Lemaitre et al., 2017], either by reducing the size of the majority class (undersampling) or by augmenting the minority class with synthetic data (oversampling). Their key advantage is compatibility with off-the-shelf machine learning algorithms and standard validation strategies, such as cross-validation. While undersampling can reduce computational complexity in training well-specified parametric models [Wang, 2020, Chen et al., 2025], it risks discarding valuable information. Oversampling, however, seeks to mitigate imbalance by generating new synthetic examples for the minority class.

The seminal oversampling algorithm, known as SMOTE (Synthetic Minority Oversampling Technique) [Chawla et al., 2002], generates new samples by interpolating between a minority instance and its nearest minority class neighbors. A different proposal called Kernel Density Estimation-based

Oversampling (KDEO) was further introduced in Gao et al. [2012], Kamalov [2020]. The present paper studies, from a theoretical perspective, the use of SMOTE and KDEO for imbalanced binary classification problems. Both methods are nonparametric and iteratively generate synthetic data points from the minority class to achieve a more balanced dataset. Similar methods such as GSMOTE [Sakho et al., 2024] can be handled using our approach, but significantly different extensions of SMOTE such as BORDERLINE-SMOTE [Han et al., 2005] and ADASYN [He et al., 2008] cannot be.

We adopt a transfer learning viewpoint, interpreting oversampling as a means to generate synthetic samples for the estimation of a modified, weighted probability measure that gives more weight to the minority class. This perspective transforms the original imbalanced learning task into a balanced one and allows us to unify reweighting and resampling approaches under a common objective: approximating the same reweighted distribution. By contrast with previous studies on SMOTE-based algorithms relying on the ROC curve [Chawla et al., 2002], AUC [Sakho et al., 2024] or F1-score [Bej et al., 2021], the metric of interest here is associated with the AM-risk to account for the fact that oversampling facilitates a shift toward a more balanced risk. Our main contributions are threefold:

- **Uniform concentration inequalities for SMOTE and KDEO.** We derive uniform concentration bounds for the discrepancy between the empirical risk on synthetic data and the population risk on the minority class, over a class of uniformly bounded functions. For SMOTE, we obtain a bound of order $n_1^{-1/2} + (k/n_1)^{1/d}$, where $n_1$ is the number of minority class samples and $k$ is the number of neighbors used in SMOTE. For KDEO, a similar bound of the form $n_1^{-1/2} + h$ is established, where $h$ is the kernel bandwidth. From these inequalities, we then derive, for each oversampling method, excess risk bounds for classifiers trained on synthetically generated data, implying consistency results for the balanced risk, provided the oversampling parameters ($k$ and $h$) are sufficiently small and the hypothesis class has controlled Rademacher complexity. This supports the commonly recommended choice of $k = 5$ when using SMOTE [Chawla et al., 2002]. For KDEO, this choice of $k = 5$ corresponds to choosing $h$ of order $n_1^{-1/d}$.

- **Excess risk bounds for nonparametric classifiers with KDEO.** We analyze the performance of a kernel smoothing plug-in classifier [Audibert and Tsybakov, 2007, Devroye et al., 2013] with KDEO. We establish an upper bound on the excess risk of this classifier, revealing a variance term of order $(n_1 h^d)^{-1/2}$ and a bias term of order $h$ or $h^2$, depending on the regularity of the class-specific distributions. This highlights a trade-off involved in choosing the kernel bandwidth $h$, whose optimal choice is different from the SMOTE-default analogous choice $n_1^{-1/d}$ and depends on the regularity of the distribution. This analysis provides novel theoretical guidance for bandwidth selection in KDEO, an issue that has remained largely heuristic in practice.

- **Empirical validation.** Numerical experiments illustrate and support the theory. They demonstrate that following the well-known Scott's rule $h \propto n_1^{-1/(d+4)}$ might be highly beneficial when using nonparametric classification rules such as *kernel smoothing* or *nearest neighbors*. This improvement is less significant when using a parametric classification rule such as *logistic regression*.

**Related Work.** While the empirical effectiveness of SMOTE is widely acknowledged, its theoretical justification remains limited. The density, mean, and variance of the sampling distribution have recently been derived [Elreedy and Atiya, 2019, Elreedy et al., 2024, Sakho et al., 2024]. As far as we are aware, our work provides the first concentration inequalities for SMOTE, thereby offering a principled explanation for its success. We rely on advanced tools from nearest neighbors theory, notably Vapnik-style inequalities [Xue and Kpotufe, 2018, Jiang, 2019, Portier, 2025].

Regarding KDEO, we go beyond earlier studies by deriving not only concentration results but also excess risk bounds for the kernel smoothing plug-in algorithm of Audibert and Tsybakov [2007], Devroye et al. [2013]. Our theory builds upon concentration bounds for the $L^1$-error [Devroye, 1991] and bias-variance decompositions from Devroye and Györfi [1985], Holmström and Klemelä [1992]. We note that Tong [2013] deals with the different problem of Neyman-Pearson classification by controlling the $\| \cdot \|_\infty$-error, rather than the $L^1$-error, of KDE. We extend these results to the case of Lipschitz densities that may exhibit discontinuities at domain boundaries – a common scenario in real-world applications.

**Outline.** Section 2 introduces the mathematical background and a formal characterization of the problem. Section 3 presents the main theoretical results and Section 4 provides supporting numerical experiments. The Appendix contains all mathematical proofs and additional numerical results.

## 2 Problem setting

### 2.1 Background

Let $\mathbb{P}$ be a probability distribution on $\mathbb{R}^d \times \{0, 1\}$, and let $\mathcal{D}_n = \{(X, Y), (X_i, Y_i)_{1 \leq i \leq n}\}$ be independent and identically distributed (i.i.d.) random variables with common distribution $\mathbb{P}$. Denote by $n_1 = \sum_{i=1}^n 1_{\{Y_i=1\}}$ and $n_0 = \sum_{i=1}^n 1_{\{Y_i=0\}}$ the number of samples from the conditional distributions $\mathbb{P}_1 := \mathbb{P}(\cdot \mid Y = 1)$ and $\mathbb{P}_0 := \mathbb{P}(\cdot \mid Y = 0)$, respectively. Under class imbalance, we assume $n_1 \ll n_0$, or equivalently, $p_1 := \mathbb{P}(Y = 1) \ll 1 - p_1 = \mathbb{P}(Y = 0)$. Denote by $\{X_{1i}\}_{1 \leq i \leq n_1}$ and $\{X_{0i}\}_{1 \leq i \leq n_0}$ the features corresponding respectively to the minority and majority classes.

Let $\ell$ be a loss function, e.g. the 0-1 loss $\ell(\alpha) = 1_{\{\alpha \leq 0\}}$ or a convex surrogate such as $\ell(\alpha) = e^{-\alpha}$, used to evaluate a discriminant function $g : \mathbb{R}^d \to \mathbb{R}$. For each pair $(X, Y)$, $\ell(g(X)(2Y - 1))$ quantifies the quality of the prediction and can be used to define the (imbalanced) risk

$$R(g) := \mathbb{E}[\ell(g(X)(2Y - 1))] = p_1 \, \mathbb{E}_1[\ell(g(X))] + (1 - p_1) \, \mathbb{E}_0[\ell(-g(X))].$$

A key challenge here is that trivial classifiers that always predict the majority class ($Y = 0$) will achieve a low such risk when $p_1$ is small, even though they perform poorly on the minority class ($Y = 1$). A common approach to mitigate this issue is to consider the $\beta$-reweighted (balanced) risk

$$R_\beta(g) := \beta \, \mathbb{E}_1[\ell(g(X))] + (1 - \beta) \, \mathbb{E}_0[\ell(-g(X))],$$

where a commonly used choice is $\beta = 1/2$, corresponding to the so-called AM-risk as studied in Menon et al. [2013], Xu et al. [2020], Aghbalou et al. [2024]. While $R_\beta(g)$ can in principle be minimized by reweighting the original samples, a widely used and successful alternative is *synthetic oversampling*. We focus on SMOTE and KDEO, that expand the minority class until the empirical class prior of the *augmented* sample matches the evaluation weight $\beta$.

### 2.2 Oversampling techniques

**SMOTE.** This method generates synthetic minority samples by interpolating between randomly selected minority examples and their nearest neighbors (NN), thereby filling in sparse regions of the minority class in the feature space. At each iteration $i$, a minority class ($Y = 1$) sample $\tilde{X}_{1i}$ is drawn uniformly at random from the minority class points $\{X_{1i}\}_{1 \leq i \leq n_1}$, and then another point $\overline{X}_{1i}$ is generated uniformly at random among the $k$NN of $\tilde{X}_{1i}$ in the minority class deprived of $\tilde{X}_{1i}$, that is, $\{X_{1j}\}_{1 \leq j \leq n_1} \setminus \{\tilde{X}_{1i}\}$ (if $n_1 = 1$, we set $\overline{X}_{1i} = \tilde{X}_{1i}$). A synthetic sample is generated as

$$X_{1i}^* = (1 - \lambda)\tilde{X}_{1i} + \lambda \overline{X}_{1i}, \tag{1}$$

where $\lambda \sim \mathcal{U}[0, 1]$ is independently drawn. The distribution of synthetic SMOTE points is then the mixture $(1/n_1) \sum_{i=1}^{n_1} \mathcal{U}_k(X_{1i})$, where $\mathcal{U}_k(X_{1i})$ denotes the (conditional) uniform distribution over the union of segments $\bigcup_{j=1}^k (X_{1i}, \hat{X}_{1j}^{(i)})$ with $\{\hat{X}_{11}^{(i)}, \ldots, \hat{X}_{1k}^{(i)}\}$ being the set of $k$ nearest neighbors of $X_{1i}$ within the minority class deprived of $X_{1i}$.

**KDEO.** This method generates synthetic minority samples by perturbing randomly selected minority points using kernel-based noise. There are again two steps. First, a point $\tilde{X}_{1i}$ is drawn uniformly at random from the minority class $\{X_{1i}\}_{1 \leq i \leq n_1}$; then, a synthetic sample is generated as

$$X_{1i}^* = \tilde{X}_{1i} + hW_i, \tag{2}$$

where $W_i \sim K$ is independently drawn from a fixed kernel distribution (e.g., standard Gaussian), and $h > 0$ is a bandwidth parameter controlling the magnitude of the perturbation. This procedure can be interpreted as, given a kernel function $K$ and bandwidth $h > 0$, drawing synthetic samples $X_{1i}^*$ from the KDE $\hat{f}_{1h}(x) = (1/n_1) \sum_{i=1}^{n_1} K_h(x - X_{1i})$ (with $K_h(x) = h^{-d} K(x/h)$) in the minority class. The key difference between SMOTE and KDEO lies in the support and shape of the component distributions: SMOTE uses uniform distributions over segments joining neighbors, while KDEO employs symmetric kernels that may perturb points in any direction.

The number $m$ of synthetic replications is chosen so that the proportion between the synthetic minority samples and the majority samples is such that $m/(m + n_0) \approx \beta$. By considering $m = n_0$, we focus on the minimization of the AM-risk $R_{1/2}$, but our results also hold for arbitrary $\beta \in (0, 1)$.

## 3 Main results

### 3.1 Concentration inequalities and excess risk bound for empirical risk minimization with oversampling

We show first that the empirical mean based on SMOTE or KDEO-generated samples concentrates uniformly around the population mean over a class of uniformly bounded functions. For SMOTE, we put a mild condition on the minority class distribution $\mathbb{P}_1$ to ensure that it puts enough mass everywhere on its support. Let $B(x, r)$ be the Euclidean ball with center $x$ and radius $r > 0$ in $\mathbb{R}^d$.

**Assumption 1.** *There exists a constant $C_d > 0$ such that for all $x \in supp(\mathbb{P}_1)$ and all $r > 0$, we have $\mathbb{P}_1(B(x, r)) \geq \min\{C_d r^d, 1\}$. [In particular, $supp(\mathbb{P}_1)$ is bounded.]*

The above assumption is standard to obtain error bounds for nearest neighbors estimators [Gadat et al., 2016, Jiang, 2019, Portier, 2025] or other local averaging methods such as local polynomial estimators [Audibert and Tsybakov, 2007]. The next assumption, not actually specific to SMOTE, introduces the class of functions over which the uniform convergence bound is derived.

**Assumption 2.** *Let $\mathcal{F}$ be a separable class of functions $G : \mathbb{R}^d \to \mathbb{R}$ which is uniformly bounded, i.e. there is $B > 0$ such that $\sup_{G \in \mathcal{F}} \|G\|_\infty \leq B$, and each $G \in \mathcal{F}$ is $L$-Lipschitz.*

One last assumption is needed to control the complexity of the class $\mathcal{F}$. This assumption relies on the well-known notion of Rademacher complexity [Bartlett and Mendelson, 2002] recalled below.

**Definition 1** (Rademacher complexity). *Let $X_1, \ldots, X_n$ be i.i.d. from a distribution $P$ on $\mathcal{X}$, and let $\sigma_1, \ldots, \sigma_n$ be i.i.d. Rademacher random variables (i.e. $\mathbb{P}[\sigma_i = 1] = \mathbb{P}[\sigma_i = -1] = 1/2$) that are independent of the $X_i$. The empirical Rademacher complexity of $\mathcal{F}$ on $X_{1:n} = (X_1, \ldots, X_n)$ is*

$$\widehat{\mathcal{R}}_n(\mathcal{F}; X_{1:n}) := \mathbb{E}_\sigma \left[ \sup_{G \in \mathcal{F}} \frac{1}{n} \sum_{i=1}^n \sigma_i G(X_i) \right].$$

**Assumption 3** (Rademacher Bound). *There exists a sequence $\mathcal{R}_n(\mathcal{F})$ such that, for every distribution $P$ on $\mathcal{X}$ with $supp(P) \subset supp(\mathbb{P}_1) \cup supp(\mathbb{P}_0)$ and $n \geq 1$, the following common upper bound holds:*

$$\mathbb{E}_{X_{1:n} \sim P^n} \left[ \widehat{\mathcal{R}}_n(\mathcal{F}; X_{1:n}) \right] \leq \mathcal{R}_n(\mathcal{F}).$$

Note that the SMOTE algorithm only applies when $n_1 > 0$. Define

$$\mu^*_{\text{Smote}}(G) := \left( \frac{1}{m} \sum_{i=1}^m G(X^*_{1i}) \right) 1_{\{n_1 > 0\}} + 0 \times 1_{\{n_1 = 0\}}.$$

While this specification is necessary to make our statement mathematically correct, we note that, in the case $n_1 = 0$, the corresponding behavior could have been specified in an arbitrary manner. We further introduce $\sigma_1^2(\mathcal{F}) = \sup_{G \in \mathcal{F}} \text{Var}(G(X)|Y = 1)$, where $X$ denotes the original data, and, likewise, we let $\hat{\sigma}_1^2(\mathcal{F}) = \sup_{G \in \mathcal{F}} \text{Var}(G(\tilde{X}_{11}) \mid \mathcal{D}_n)$ given that $n_1 > 0$, where $\tilde{X}_{11}$ is drawn uniformly at random from the minority class $\{X_{1i}\}_{1 \leq i \leq n_1}$. We make below the convention $\mathcal{R}_0(\mathcal{F}) = 1/0 = +\infty$ and we set $k_\delta = \max(k, (d+1) \log(2n_1) + \log(8/\delta))$ on the event $\{n_1 > 1\}$ and 0 otherwise.

**Theorem 1.** *Suppose that Assumptions 1, 2 and 3 hold. Let $\delta \in (0, 1/5)$ and $m \geq 1$. If $n_1 > 1$, let $k \in \{1, \ldots, n_1 - 1\}$ and let $\{X^*_{1i}\}_{1 \leq i \leq m}$ be $m$ i.i.d. samples generated by the SMOTE algorithm (1). Then, with probability at least $1 - 5\delta$,*

$$\sup_{G \in \mathcal{F}} |\mu^*_{\text{Smote}}(G) - \mathbb{E}_1[G(X)]| \leq 4\mathcal{R}_m(\mathcal{F}) + 4\mathcal{R}_{n_1}(\mathcal{F}) + L \left( \frac{6}{C_d} \right)^{1/d} \left( \frac{k_\delta}{n_1} \right)^{1/d} + R,$$

*where*

$$R = \left( \sqrt{\frac{\hat{\sigma}_1^2(\mathcal{F})}{m}} + \sqrt{\frac{\sigma_1^2(\mathcal{F})}{n_1}} \right) \sqrt{2 \log \left( \frac{1}{\delta} \right)} + \frac{8}{3} B \left( \frac{1}{m} + \frac{1}{n_1} \right) \log \left( \frac{1}{\delta} \right).$$

We apply this to the proof of an excess risk bound for empirical risk minimization algorithms under synthetic SMOTE-based oversampling. Let $\mathcal{G}$ be a class of real-valued discriminative functions

defined on $\mathbb{R}^d$ and let $\ell : \mathbb{R} \to [0, \infty)$ be the loss function. Consider the following classifier:

$$\hat{g}^*_{\mathcal{G}} \in \operatorname{argmin}_{g \in \mathcal{G}} \left\{ 1_{\{n_1 > 0\}} \sum_{i=1}^{m} \ell(g(X^*_{1i})) + 1_{\{n_0 > 0\}} \sum_{i=1}^{n_0} \ell(-g(X_{0i})) \right\}. \tag{3}$$

This is a standard classifier, using SMOTE-generated data. By convention, and similarly to what was done with $\mu^*_{\text{Smote}}(G)$ above, the first term (resp. second term) in the above minimization is set as 0 when $n_1 = 0$ (resp. $n_0 = 0$). Let $\ell(\mathcal{G}) = \{\ell \circ g, \ g \in \mathcal{G}\}$ and $\sigma_0^2(\mathcal{F}) = \sup_{G \in \mathcal{F}} \operatorname{Var}[G(X)|Y=0]$.

**Corollary 2.** *Let $m = n_0 1\{n_1 > 0\}$ and $\delta \in (0, 1/7)$. Under the assumptions of Theorem 1 with $\mathcal{F} = \ell(\mathcal{G}) \cup \ell(-\mathcal{G})$, we have, with probability at least $1 - 7\delta$,*

$$R_{1/2}(\hat{g}^*_{\mathcal{G}}) - \inf_{g \in \mathcal{G}} R_{1/2}(g) \le 4\mathcal{R}_{n_1}(\mathcal{F}) + 8\mathcal{R}_{n_0}(\mathcal{F}) + L \left( \frac{6}{C_d} \right)^{1/d} \left( \frac{k_\delta}{n_1} \right)^{1/d} + R,$$

*where*

$$R = \left( \sqrt{\frac{\hat{\sigma}_1^2(\mathcal{F})}{n_0}} + \sqrt{\frac{\sigma_1^2(\mathcal{F})}{n_1}} + \sqrt{\frac{\sigma_0^2(\mathcal{F})}{n_0}} \right) \sqrt{2 \log \left( \frac{1}{\delta} \right)} + \frac{8}{3} B \left( \frac{2}{n_0} + \frac{1}{n_1} \right) \log \left( \frac{1}{\delta} \right).$$

We turn to KDEO. For $q > 0$ and any measurable $f : \mathbb{R}^d \to [0, \infty)$, let $M_q(f) := \int \|z\|_2^q f(z) dz$. Define $\mu^*_{\text{KDEO}}(G)$ as $\mu^*_{\text{Smote}}(G)$, only with synthetic data generated by KDEO rather than SMOTE.

**Theorem 3.** *Suppose that Assumptions 2 and 3 hold. Let $K$ be a density function with $M_1(K) < \infty$. Let $\delta \in (0, 1/5)$ and $m \ge 1$. Moreover, whenever $n_1 > 0$, let $\{X^*_{1i}\}_{1 \le i \le m}$ be $m$ i.i.d. samples generated by the KDEO algorithm (2). Then, with probability at least $1 - 5\delta$,*

$$\sup_{G \in \mathcal{F}} |\mu^*_{\text{KDEO}}(G) - \mathbb{E}_1[G(X)]| \le 4\mathcal{R}_{n_1}(\mathcal{F}) + 4\mathcal{R}_m(\mathcal{F}) + 5LhM_1(K) + R,$$

*where*

$$R = \left( \sqrt{\frac{\sigma_1^2(\mathcal{F})}{n_1}} + \sqrt{\frac{\hat{\sigma}_1^2(\mathcal{F})}{m}} \right) \sqrt{2 \log \left( \frac{1}{\delta} \right)} + \frac{8}{3} B \left( \frac{27}{8m} + \frac{1}{n_1} \right) \log \left( \frac{1}{\delta} \right).$$

For the classifier defined in (3) with KDEO and not SMOTE, we obtain the excess risk bound below.

**Corollary 4.** *Let $m = n_0 1\{n_1 > 0\}$ and $\delta \in (0, 1/7)$. Under the assumptions of Theorem 3 with $\mathcal{F} = \ell(\mathcal{G}) \cup \ell(-\mathcal{G})$, we have, with probability at least $1 - 7\delta$,*

$$R_{1/2}(\hat{g}^*_{\mathcal{G}}) - \inf_{g \in \mathcal{G}} R_{1/2}(g) \le 4\mathcal{R}_{n_1}(\mathcal{F}) + 8\mathcal{R}_{n_0}(\mathcal{F}) + 5LhM_1(K) + R,$$

*where*

$$R = \left( \sqrt{\frac{\hat{\sigma}_1^2(\mathcal{F})}{n_0}} + \sqrt{\frac{\sigma_1^2(\mathcal{F})}{n_1}} + \sqrt{\frac{\sigma_0^2(\mathcal{F})}{n_0}} \right) \sqrt{2 \log \left( \frac{1}{\delta} \right)} + \frac{8}{3} B \left( \frac{35}{8n_0} + \frac{1}{n_1} \right) \log \left( \frac{1}{\delta} \right).$$

**Remark 1.** *The Lipschitz property required in Corollaries 2 and 4 as part of Assumption 2 holds for standard loss functions such as hinge and logistic, and for classifiers like neural networks under mild conditions such as bounded spectral norms of the weight matrices.*

**Remark 2.** *Many practical function classes satisfy the Rademacher complexity bound in Assumption 3 under natural constraints, with $\mathcal{R}_n(\mathcal{F}) \le B_{\mathcal{F}}/\sqrt{n}$, where $B_{\mathcal{F}} > 0$ is a constant. For example, if $\mathcal{F}$ has finite VC–subgraph dimension $v$, then the previous bound is valid with $B_{\mathcal{F}}$ depending on $v$, $B$ and $\sigma_1^2(\mathcal{F})$ [Giné and Guillou, 2001, Proposition 2.1]; for linear predictors based on $\mathcal{F} = \{x \mapsto w^\top x : \|w\|_2 \le W\}$ with $\|x\|_2 \le R$, one has that $B_{\mathcal{F}}$ depends on $R$ and $W$ and the ambient dimension [Bartlett and Mendelson, 2002, Lemma 19]; and for fully-connected ReLU networks of depth $L$ whose inputs satisfy $\|x\|_2 \le R$ and whose weight matrices $A_\ell$ obey the Frobenius-norm bounds $\|A_\ell\|_F \le s_\ell$, $B_{\mathcal{F}}$ scales as $R\sqrt{L} \prod_{\ell=1}^{L} s_\ell$ [Golowich et al., 2018, Theorem 1].*

**Remark 3.** *The bounds of Corollaries 2 and 4 depend on the sample sizes $n_0$ and $n_1$ which, in our framework, are random variables. Considering $n_1$, a multiplicative Chernoff bound gives that with probability at least $1 - \delta$, $n_1 \ge np_1(1 - \epsilon)$, where $\epsilon^2 = 2 \log(1/\delta)/(np_1)$. Doing so reveals a frontier which is, in the asymptotic regime, $np_1 \to \infty$. When $p_1$ is below this frontier, it is not clear that consistent estimation is possible as the bound does not vanish asymptotically.*

### 3.2  Nonparametric excess risk bound for the KDEO-based kernel smoothing plug-in classifier

The balanced Bayes classifier minimizing the AM-risk is $g(x) = 1_{\{\eta(x) > p_1\}}$, where $\eta(x) = \mathbb{P}(Y = 1 | X = x)$. Besides, if $\mathbb{P}_y$ has a density function $f_y$ with respect to the Lebesgue measure then, by the law of total probability, the distribution of $X$ has density $f = p_1 f_1 + (1 - p_1) f_0$ with respect to this same measure. It follows from the Bayes formula that the balanced Bayes classifier is then given by

$$\forall x \in \mathbb{R}^d, \quad g(x) = 1_{\{f_1(x) > f(x)\}} = 1_{\{f_1(x) > f_0(x)\}}. \tag{4}$$

Our ultimate goal is to study the counterpart of $g$ using a KDEO-based kernel smoothing classifier. Before that, we give a general result, with respect to the AM-risk, on the performance of any discrimination rule $\hat{g}$ of the form $\hat{g}(x) = 1_{\{\hat{f}_1(x) > \hat{f}_0(x)\}}$, where each $\hat{f}_y$ is an estimator of $f_y$.

**Theorem 5.** *For each $y \in \{0, 1\}$, suppose that $\mathbb{P}_y$ has a density $f_y$ with respect to the Lebesgue measure. Let $S_y$ denote the support of $\mathbb{P}_y$ and $S = S_0 \cup S_1$. We have*

$$R_{1/2}(\hat{g}) - R_{1/2}(g) \leq \frac{1}{2} \int_S |\hat{f}_1(x) - f_1(x)| dx + \frac{1}{2} \int_S |\hat{f}_0(x) - f_0(x)| dx.$$

This is reminiscent of known results in [Devroye et al., 2013, Chapter 6] or [Devroye and Györfi, 1985, Chapter 10], showing that the excess classification risk is bounded by the $L^1$-error of the conditional probability estimators or the density estimators in the different classes. Theorem 5, though, is concerned with the AM-risk while the above references deal with the standard risk measure.

Let us now focus on the kernel smoothing classifier of Audibert and Tsybakov [2007], Devroye et al. [2013] when using KDE-generated data. Consider, as in (2), independent samples $\{X^*_{1i}\}_{1 \leq i \leq m}$ with common density $\hat{f}_{1h}$ (given the initial sample), i.e. the KDE of the minority class covariates. Define

$$\hat{\eta}^*(x) := \frac{\sum_{i=1}^m K_s(x - X^*_{1i})}{\sum_{i=1}^m K_s(x - X^*_{1i}) + \sum_{i=1}^{n_0} K_s(x - X_{0i})} = \frac{m \hat{f}^*_{1s}(x)}{m \hat{f}^*_{1s}(x) + n_0 \hat{f}_{0s}(x)},$$

with the convention $0/0 = 0$ and where $\hat{f}^*_{1s}(x) = 1_{\{n_1 > 0\}}(1/m) \sum_{i=1}^m K_s(x - X^*_{1i})$ is the kernel density estimate based on synthetic data from class 1 and $\hat{f}_{0s}$ is the kernel density estimate based on initial data from class 0 (with $\hat{f}_{0s} = 0$ when $n_0 = 0$). One could, of course, choose a different kernel for $\hat{f}_{0s}$, $\hat{f}^*_{1s}$ and $\hat{f}_{1h}$; we take the same kernel $K$ in each estimate for the sake of simplicity.

By setting $m = n_0 1_{\{n_1 > 0\}}$, the discrimination rule $\hat{g}^*(x) = 1_{\{\hat{\eta}^*(x) > 1/2\}} = 1_{\{\hat{f}^*_{1s}(x) > \hat{f}_{0s}(x)\}}$ is tailored to minimizing the AM-risk. Next we obtain an upper bound on the excess risk of $\hat{g}^*$ in this setting. Regularity assumptions are required; let, for any open subset $U$ of $\mathbb{R}^d$, $W^{s,1}(U)$ be the Sobolev space of functions $G : U \to \mathbb{R}$ whose (weak) partial derivatives of order $s$ are integrable.

**Assumption 4.** $K : \mathbb{R}^d \to \mathbb{R}$ *is a square-integrable symmetric density function such that $M_{d+\varepsilon}(K + K^2) < \infty$ for some $\varepsilon > 0$ and $M_2(K) < \infty$.*

**Assumption 5.** *For each $y \in \{0, 1\}$, $\mathbb{P}_y$ has a density $f_y \in W^{2,1}(\mathbb{R}^d)$ with respect to the Lebesgue measure and $M_{d+\varepsilon}(f_y) < \infty$ for some $\varepsilon > 0$.*

To deal with situations violating Assumption 5, where the densities may, for example, have compact supports and be smooth and bounded away from zero on the interior of their supports, we introduce alternative assumptions. For two sets $S_1$ and $S_2$, $S_1 + S_2$ is the set $\{s_1 + s_2, (s_1, s_2) \in S_1 \times S_2\}$. The Lebesgue measure is denoted by $\lambda$ and the topological boundary of a set $E$ is denoted by $\partial E$.

**Assumption 6.** $K : \mathbb{R}^d \to \mathbb{R}$ *is a square-integrable symmetric density function supported on $B(0, 1)$.*

**Assumption 7.** *For each $y \in \{0, 1\}$, the support $S_y$ of $\mathbb{P}_y$ is smooth, in the sense that there are $\kappa, r_0 > 0$ such that $\lambda(\partial S_y + B(0, r)) \leq \kappa r$ for $0 < r < r_0$. Moreover, $\mathbb{P}_y$ has a density $f_y \in W^{1,1}(U)$ with respect to the Lebesgue measure, where $U$ is the interior of $S$, which is bounded on $\partial S + B(0, 2r_0)$, and $M_{d+\varepsilon}(f_y) < \infty$ for some $\varepsilon > 0$.*

Assumption 7 on $S_y$ is, for example, satisfied if $\partial S_y$ is a finite union of compact, closed and smooth submanifolds of dimension $d - 1$ in $\mathbb{R}^d$ whose pairwise distances to one another are nonzero. This is a consequence of Weyl's tube formula in the Euclidean space, see Equation (14) in Weyl [1939] applied to manifolds of codimension $m = 1$ with the notation therein; a self-contained statement of this result is Theorem 9.3.11 in Nicolaescu [2007].

These assumptions make it possible to prove $L^1-$bounds on the KDEs of the $f_y$. The proofs rest upon the McDiarmid inequality [McDiarmid, 1989] and a bias-variance decomposition, where it is shown and used that, if $f_y * K_h(x) = \int K_h(x-z)f_y(z)dz$ denotes the convolution of $f_y$ and $K_h$,

$$\int \sqrt{\mathbb{E}|\hat{f}_{yh}(x) - f_y * K_h(x)|^2}dx \leq c_{y,d,\varepsilon}\sqrt{\frac{M_0(K^2)}{n_y h^d}}(1 + h^{(d+\varepsilon)/2})$$

$$\text{and} \quad \int |f_y * K_h(x) - f_y(x)|dx \leq \begin{cases} \phi_y h^2 & \text{under Assumptions 4 and 5,} \\ \psi_y h & \text{if } h \leq r_0 \text{ under Assumptions 6 and 7.} \end{cases}$$

The constant $c_{y,d,\varepsilon}$ (resp. $\phi_y$, $\psi_y$), defined right below Lemma 5 (resp. Lemma 6), depends on $d$, $\varepsilon$, $f_y$ and $K$ (resp. $f_y$ and $K$) only, see the Appendix. Let us introduce $\hat{c}_y$, obtained from $c_{y,d,\varepsilon}$ by plugging in $\hat{f}_{yh}$ instead of $f_y$. One may likewise obtain a (conditional) $L^1-$bound on the KDE $\hat{f}_{1s}^*$ based upon the oversampled covariates $\{X_{1i}^*\}_{1 \leq i \leq m}$ in the minority class. Combining these bounds with Theorem 5 results in the following bound on the excess risk of the classification rule $\hat{g}^*$.

**Theorem 6.** *Let $m = n_0 1_{\{n_1 > 0\}}$ and $\delta \in (0, 1/3)$. Then, with probability at least $1 - 3\delta$,*

$$R_{1/2}(\hat{g}^*) - R_{1/2}(g)$$
$$\leq \frac{\sqrt{M_0(K^2)}}{2}\left(c_{0,d,\varepsilon}\sqrt{\frac{1}{n_0 s^d}} + c_{1,d,\varepsilon}\sqrt{\frac{1}{n_1 h^d}} + \hat{c}_{1,d,\varepsilon}\sqrt{\frac{1}{n_0 s^d}}\right)(1 + \max(h,s)^{(d+\varepsilon)/2})$$
$$+ \sqrt{2\log\left(\frac{1}{\delta}\right)}\left(\frac{1}{\sqrt{n_0}} + \frac{1}{2\sqrt{n_1}}\right)$$
$$+ \frac{1}{2}\begin{cases} \phi_0 s^2 + \phi_1(h^2 + s^2) & \text{under Assumptions 4 and 5,} \\ \psi_0 s + \psi_1(h + s) & \text{if } h, s \leq r_0 \text{ under Assumptions 6 and 7.} \end{cases}$$

In Theorem 6 the leading term (in case of second order regularity) is scaling as $(n_0 s^d)^{-1/2} + s^2 + (n_1 h^d)^{-1/2} + h^2$. The optimal value of the second bandwidth $s$, involved in the error of $\hat{f}_{0s}$ and $\hat{f}_{1s}^*$ (based on $n_0$ observations), is then $s = n_0^{-1/(d+4)}$, while the optimal value for the first bandwidth $h$, involved in the data generation step (based on $n_1$ observations) is $h = n_1^{-1/(d+4)}$. In highly imbalanced scenarios where $n_1/n_0 \to 0$, the two bandwidths should thus be set differently to optimize the upper bound and reach the optimal convergence rate $n_0^{-2/(d+4)} + n_1^{-2/(d+4)}$.

**Remark 4** (Comparison with SMOTE). *The default value in SMOTE is $k = 5$ [Chawla et al., 2002] which, viewing kNN as a kernel estimator with data-driven bandwidth (see e.g. Portier et al. [2024], Lemma 3), would correspond to a choice of $h \simeq (1/n_1)^{1/d}$ (omitting constants). This is different from the optimal choice above, and it would not (based on our upper bound) guarantee consistency, as the upper bound in Theorem 6 would not converge to 0.*

**Remark 5** (Comparison with the kernel smoothing plug-in rule). *Our method of proof allows to establish an excess AM-risk bound for the kernel smoothing rule $\hat{g}_h(x) = 1_{\{\hat{f}_{1h}(x) > \hat{f}_{0h}(x)\}}$ based on the initial data only (see Proposition 9 in the Appendix). The bound scales as $(n_1 h^d)^{-1/2} + h^2 + (n_0 h^d)^{-1/2} + h^2$, the optimal value for $h$ being $h = (1/n_1 + 1/n_2)^{1/(d+2)}$. The bound obtained in Theorem 6 is similar but different, as the two bandwidth parameters, $h$ and $s$, might be set differently.*

These remarks suggest that the synthetic oversampling rule $\hat{g}^*$ (i) performs no worse than the rule $\hat{g}_h$ based on the initial data only, (ii) should be expected to perform better than a rule using SMOTE synthetic generation with default parameters, and (iii) improvements may be observed in practice by choosing carefully $s$. In the numerical experiments, by considering the $K$-NN classifier, which can be seen as a practical modification of the kernel smoothing classifier, we investigate a cross-validation procedure to choose $K$ (having similar role as $s$), while $h$ is chosen following Scott's rule.

Investigating the $K$NN plug-in rule (instead of kernel smoothing) in Theorem 6 as well as fast convergence rates under the noise condition of Audibert and Tsybakov [2007], Tong [2013] remain open problems. Finally, Theorem 6 is only valid for KDE-based sampling and might be extended to SMOTE. None of these problems are direct consequences of this work.

# 4 Numerical experiments

## 4.1 Methods in competition

**Oversampling techniques.** The SMOTE and KDEO methods are applied to imbalanced data with $n_1 < n_0$. After oversampling, the synthetic data will contain $m = n_0$ observations with label 1 and $n_0$ observations with label 0. For SMOTE, we consider the default choice $k = 5$ neighbors. For KDEO, we consider the matrix-valued bandwidth $H_1$ such that $H_1^2 = n_1^{-2/(d+4)}C_1$, following from Scott's rule, where $C_1$ is the covariance matrix computed from the minority class samples.

**Classification methods.** We consider the kernel smoothing (KS) discrimination rule studied in Section 3.2 and the $K$NN classification rule as follows. First, apply the concerned oversampling technique, either SMOTE or KDEO, and then employ the $K$NN (resp. KS) algorithm with parameter $K = \sqrt{n}$ (resp. $s = S_j$ obtained from Scott's rule $S_j^2 = n_0^{-2/(d+4)}C_j$, $j = 0, 1$, where $C_j$ is the covariance matrix of class $j$). This choice of $S_j$ (as well as $H_1$ for KDEO) corresponds to the optimal scaling recommended by Theorem 6. Note that both $K$NN and KS are based on the local averaging principle. This is compared to the logistic regression (LR) method, employed to incorporate a parametric classification approach and to illustrate the theory developed in Section 3.1. Finally, we also consider the $K$-NN balanced Bayes classifier (BBC), which does not involve oversampling but rather reweighting. It is defined as the classifier $1_{\{\hat{\eta}_{NN}(x) > \hat{p}\}}$ where $\hat{\eta}_{NN}(x)$ is the $K$NN estimator of $\eta(x)$ with hyperparameter $K = \sqrt{n}$ and $\hat{p} = n_1/n$. Under class imbalance, the threshold adjustment allows to minimize the AM-risk [Aghbalou et al., 2024]. Similar results with a Random Forest classifier applied after SMOTE and KDEO are given in the Appendix.

## 4.2 Simulated data

Four models are considered. In all cases, $\{(X_i, Y_i)\}_{1 \le i \le n}$ are $n = 1000$ i.i.d. training samples from the distribution of $(X, Y)$. Let $e_i$ be the $i$th vector in the canonical basis of $\mathbb{R}^d$.

- **Example 1:** Let $X \sim \mathcal{N}(0, I_d)$ and $Y \sim$ Bernoulli$(\text{expit}(X^\top e_1 + \alpha))$, with $\alpha$ tuning class imbalance and $\text{expit}(u) = \exp(u)/(1 + \exp(u))$.

- **Example 2:** Let $X \sim \mathcal{N}(0, I_d)$ and $Z \sim$ GPD$(\sigma(X), \xi = 0.5)$ have a Generalized Pareto distribution, where $\sigma(X) = \exp(X^\top e_1)$. Define $Y = 1_{\{Z > t\}}$ with $t$ tuning class imbalance.

- **Example 3:** Let $Z = \mathcal{B}\sin(X^\top e_1/2)Y_1 + (1 - \mathcal{B})\sin(X^\top e_2/2)Y_2$, with $\mathcal{B} \sim$ Bernoulli$(0.5)$, $X \sim \mathcal{N}(0, I_d)$, $Y_1 \sim$ GPD$(1, 0.5)$, $Y_2 \sim$ Exp$(10)$. Set $Y = 1_{\{Z > t\}}$ with $t$ tuning class imbalance.

- **Example 4:** Let $d = 2$ and $\mu_1 = (0, 0)^\top$, $\mu_2 = (10, 10)^\top$, $\mu_3 = (10, 0)^\top$, $\mu_4 = (0, 10)^\top$. Let $Z$ be $\{1, 2, 3, 4\}$-valued with $\mathbb{P}(Z = c) = w_c$, then $X \sim \sum_{c=1}^4 \mathcal{N}(\mu_c, 6I_p)1_{\{Z=c\}}$ and $Y = 1_{\{Z \ge 3\}}$.

Take $d = 4$ in Examples 1, 2 and 3. In each case, the validation sample is created by generating 10000 observations and then undersampling the majority class so as to obtain a balanced data set.

**Results when varying $k$ in SMOTE and $H$ in KDEO.** In Figure 1, different values of $k$ and $H$ are considered when dealing with Examples 2 and 3 for which the binary response $Y$ was constructed by thresholding at the probability $1 - p_1 = 0.90$ (for Examples 1 and 4, see the Appendix). We consider SMOTE$(k)$ with varying $k \in (7, 65)$, and KDEO$(H)$ with varying $H = cH_1$, for $c$ ranging in $(1/20, 3)$, where $H_1$ follows from Scott's rule. The KS, $K$NN and LR classification methods are considered. Figure 1 displays the average (over 50 replications) AM-risk over the validation set.

In Figure 1, we see that varying $k$ in SMOTE$(k)$ or the bandwidth $H$ in KDEO$(H)$ leads to noticeable changes in the AM-risk values. In contrast, when using LR, changing $H$ or $k$ has almost no effect. This indicates that LR is considerably less sensitive to the choice of oversampling parameters $k$ or $H$ compared to nonparametric classifiers. This was already suggested by Corollaries 2 and 4, while the impact of a real-valued bandwidth was formally analyzed in Theorem 6 when using KS. Observe also that, in Figure 1 (left panel), the optimal value of $k$ for SMOTE(k) when using $K$NN lies between 45 and 60. This differs substantially from the default $k = 5$. Figure 1 (right panel) shows that varying the bandwidth $H$ does not yield improvements over KDEO, which relies on Scott's rule of thumb, that in fact turns out to be optimal. Finally, note that KDEO$(H)$ with small $H$ produces results that closely resemble those of SMOTE, which is consistent with the intuition that $k$ and $H$ play a similar role.

**Results under different degrees of class imbalance.** The methods are evaluated on Examples 2 and 3 (for Examples 1 and 4, see the Appendix), where the parameters $t$, $\alpha$ and $w_c$ are adjusted to achieve different probability levels ($1 - p_1 = 0.60, 0.80, 0.85, 0.90, 0.92, 0.95$). The AM-risk of the $K$NN and KS classifiers for Examples 2 and 3 is reported in Figure 2 under KDEO and SMOTE.

Across all examples, both classifiers exhibit similar patterns. A clear performance gain is observed for KDEO and BBC compared with SMOTE. In particular, the AM risk of the KS classifier under KDEO sampling is consistently smaller than that of $K$NN when trained under SMOTE. These results suggest that fine-tuning the bandwidth $H_1$ in KDEO, together with the use of the KS classifier based on well-chosen matrices $S_j$, provides clear benefits. This, in turn, opens the perspective of tuning $K$NN using cross-validation, as investigated below. By contrast, the LR classifier performs uniformly across all resampling methods (see the Appendix). This is in line with the results given in Theorems 1 and 3 as they suggest that estimating the risk with SMOTE or KDEO gives better results when $K$ and $H$ are small, underlining that oversampling may not be critical for such a parametric classifier.

**Cross validation (CV) selection of $K$ in $K$NN: SMOTE-CV, KDEO-CV, BBC-CV.** To improve the $K$NN classifier's performance, we consider selecting $K$ via cross-validation. Specifically, we use 5-fold cross-validation, where in each fold, the data is first balanced using the chosen oversampling method (SMOTE/KDEO), and then $K$ is selected (from a grid) to minimize the validation error. For the BBC, the procedure differs: we apply the BBC classifier obtained from training folds and choose the value of $K$ that minimizes an AM-risk estimate obtained via under-sampling of the testing fold.

The AM-risk across BBC, SMOTE, and KDEO, and the CV choice of $K$ are reported for Examples 2 and 3 in Figure 3 (resp. Examples 1 and 4 given in the Appendix). A noticeable improvement can be observed in the results for KDEO-CV and SMOTE-CV. The CV on $K$ yields poorer performance for the BBC, suggesting that the BBC and CV may not interact effectively in this context. Moreover, KDEO and KDEO-CV consistently outperform SMOTE, SMOTE-CV, and BBC, even in scenarios where the probability $p_1$ is smaller. The intuition is that KDEO, which generates synthetic samples using kernel density estimates, may better approximate the true minority distribution than SMOTE.

### 4.3 Real data analysis: Abalone, California, MagicTel, Phoneme, and House_16H datasets

Each dataset is split into training and validation sets using a $70 : 30$ ratio. California and MagicTel are balanced datasets; to make the training set imbalanced, we subsample the minority class in these two datasets, adjusting the imbalance ratio to $20\%$, $10\%$, and $5\%$. The validation sets are balanced by undersampling the majority class to ensure fair assessment aligned with the evaluation metric.

The AM-risk results for the $K$NN classifier are presented in Figure 4 (for alternative classifiers, see the Appendix). SMOTE and KDEO generally perform similarly except for Abalone, where SMOTE consistently achieves a smaller risk. This suggests that the Abalone features might have a specific structure that is better synthesized by nearest neighbors rather than by the (more arbitrary) Gaussian kernel. One particularly relevant conclusion is that using CV for choosing $K$ compared to the choice $K = \sqrt{n}$ (almost) continuously improves the performance across all the different cases. A similar pattern is observed for Random Forest, LR, and LR-Lasso, see the Appendix.

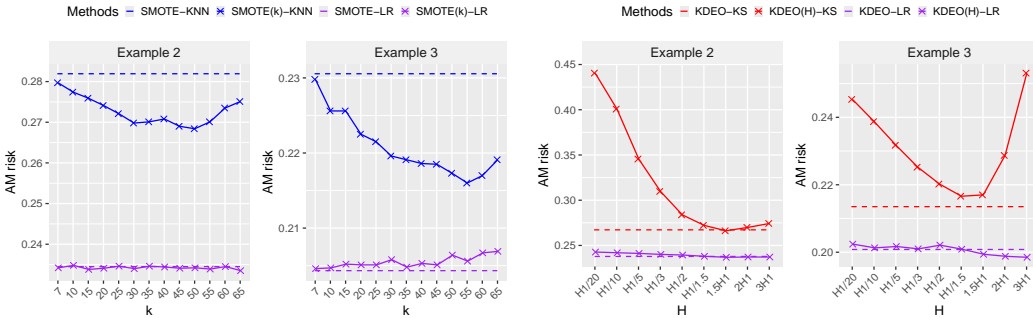

Figure 1: Average AM-risk of KNN, KS, and LR classifiers on balanced data over 50 replications. *Left:* using SMOTE and SMOTE($k$) with $k \in (7, 65)$. *Right:* using KDEO and KDEO($H$), with $H = cH_1$ and $c$ ranging in $(1/20, 3)$ where $H_1$ follows from Scott's rule.

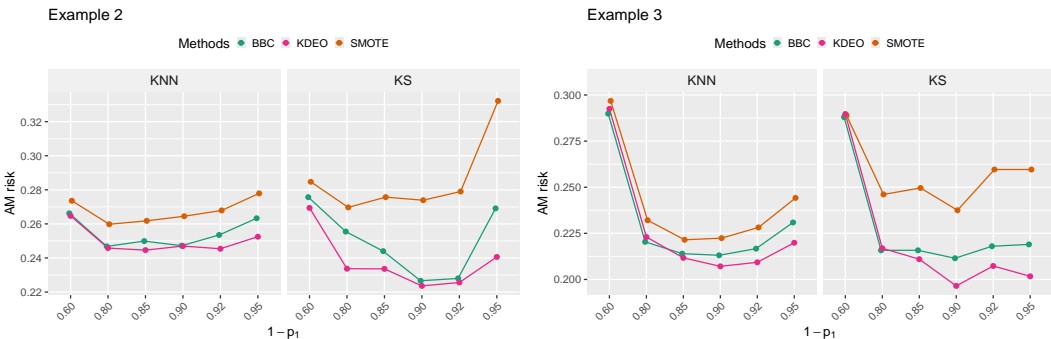

Figure 2: Average AM-risk across different data imbalance regimes for the KS (described in Section 3.2) and $K$NN classification rules computed over 50 replications.

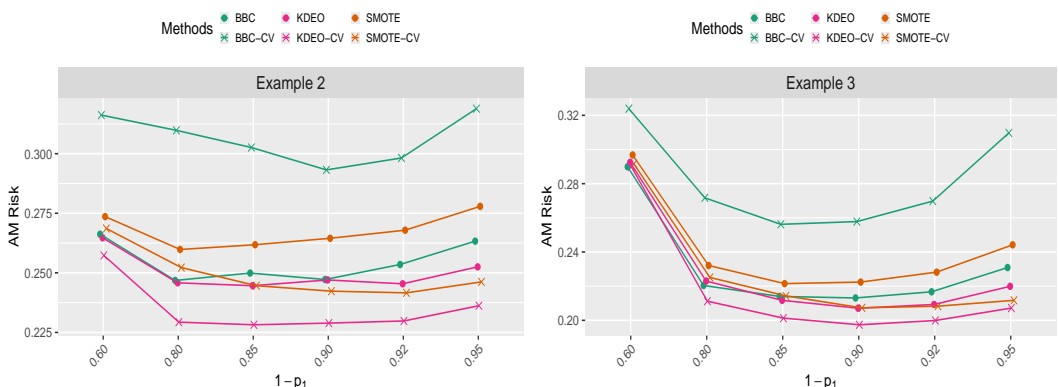

Figure 3: Average AM-risk across different data imbalance regimes for the $K$NN methods described in Section 4.1, computed over 50 replications.

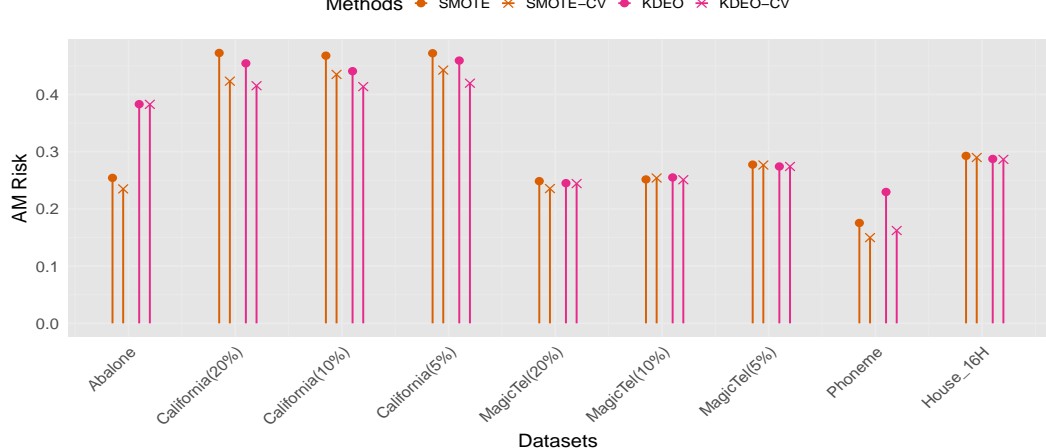

Figure 4: AM-risk corresponding to different rebalancing methods and datasets when using the $K$NN classifier.

## Acknowledgments

T. Ahmad acknowledges support from the Région Bretagne through project SAD-2021-MaEVa. G. Stupfler acknowledges support from grants ANR-19-CE40-0013 (ExtremReg project), ANR-23-CE40-0009 (EXSTA project) and ANR-11-LABX-0020-01 (Centre Henri Lebesgue), the TSE-HEC ACPR Chair "Regulation and systemic risks", and the Chair Stress Test, RISK Management and Financial Steering of the Foundation Ecole Polytechnique.

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

# Appendix to "Concentration and excess risk bounds for imbalanced classification with synthetic oversampling"

Touqeer Ahmad, Mohammadreza M. Kalan, François Portier, Gilles Stupfler

This appendix is organized as follows. We provide auxiliary theoretical results in Appendix A, among which classical concentration inequalities and inequalities for convolution of density functions. We then prove Theorem 1 in Appendix B and Theorem 3 in Appendix C. A unified proof of Corollaries 2 and 4 is given in Appendix D. Appendix E and Appendix F are dedicated to the proofs of Theorems 5 and 6, respectively. Appendix G gives the statement of Proposition 9 which is used in Remark 5 of the paper. Appendix H gives a further set of numerical results complementing those of Section 4.

## A Auxiliary results

### A.1 Concentration inequalities

The following lemma provides a uniform bound on the distance between a point and its $k$-th nearest neighbor.

**Lemma 1** (Xue and Kpotufe [2018], Lemma 1). *Fix $n_1 > 0$ and let $\{X_{1i}\}_{1 \leq i \leq n_1}$ be i.i.d. samples from a distribution $\mathbb{P}_1$ satisfying Assumption 1. Define $r_k(x)$ as the Euclidean distance from a point $x \in supp(\mathbb{P}_1)$ to its $k$-th nearest neighbor in $\{X_{1i}\}_{1 \leq i \leq n_1}$. Then, for all $\delta \in (0, 1)$, with probability at least $1 - \delta$, it holds that*

$$\forall k \in \{1, \ldots, n_1\}, \quad \sup_{x \in supp(\mathbb{P}_1)} r_k(x) \leq \left(\frac{3}{C_d}\right)^{1/d} \left(\frac{\max(k, (d+1)\log(2n_1) + \log(8/\delta))}{n_1}\right)^{1/d}.$$

We next recall, without proof, the classical McDiarmid and Talagrand–Bousquet inequalities.

**Lemma 2** (McDiarmid inequality, see McDiarmid [1989]). *Let $Z_1, Z_2, \ldots, Z_n$ be independent random variables taking values in some set $\mathcal{X}$. Let $T : \mathcal{X}^n \to \mathbb{R}$ be a function satisfying the **bounded differences condition**: for each $i \in \{1, 2, \ldots, n\}$, there exists a constant $C_i \geq 0$ such that for all $z_1, \ldots, z_n, z_i' \in \mathcal{X}$,*

$$|T(z_1, \ldots, z_{i-1}, z_i, z_{i+1}, \ldots, z_n) - T(z_1, \ldots, z_{i-1}, z_i', z_{i+1}, \ldots, z_n)| \leq C_i.$$

*Then, for all $t > 0$,*

$$\mathbb{P}(T(Z_1, \ldots, Z_n) - \mathbb{E}[T(Z_1, \ldots, Z_n)] \geq t) \leq \exp\left(-\frac{2t^2}{\sum_{i=1}^n C_i^2}\right)$$

$$and \quad \mathbb{P}(T(Z_1, \ldots, Z_n) - \mathbb{E}[T(Z_1, \ldots, Z_n)] \leq -t) \leq \exp\left(-\frac{2t^2}{\sum_{i=1}^n C_i^2}\right).$$

*In particular, for any $\delta \in (0, 1/2)$, it holds with probability at least $1 - 2\delta$ that*

$$|T(Z_1, \ldots, Z_n) - \mathbb{E}[T(Z_1, \ldots, Z_n)]| \leq \sqrt{\frac{\log(1/\delta)}{2} \sum_{i=1}^n C_i^2}.$$

**Lemma 3** (Talagrand [1996], Theorem 2.3 in Bousquet [2002]; with separability assumptions, Boucheron et al. [2013] p.315). *Let $Z_1, \ldots, Z_n$ be independent and identically distributed random variables with values in a measurable space $\mathcal{X}$. Let $\mathcal{F}$ be a separable class of measurable functions $f : \mathcal{X} \to \mathbb{R}$ that satisfy, for some $U > 0$,*

$$\mathbb{E}f(Z_i) = 0 \quad and \quad |f(x)| \leq U \quad for\ all\ f \in \mathcal{F},\ x \in \mathcal{X}.$$

*Let $\sigma^2 = \sup_{f \in \mathcal{F}} \text{Var}(f(Z_1))$. Then for every $t \geq 0$,*

$$\mathbb{P}\left(S_n \geq \mathbb{E}(S_n) + \sqrt{2(n\sigma^2 + 2U\mathbb{E}(S_n))t} + \frac{U}{3}t\right) \leq e^{-t}, \tag{5}$$

*where $S_n$ can be either equal to $\sup_{f \in \mathcal{F}} \sum_{i=1}^n f(Z_i)$ or $\sup_{f \in \mathcal{F}} |\sum_{i=1}^n f(Z_i)|$.*

The next lemma simplifies the bound in the conclusion (5) of Lemma 3 to a form that is more convenient for our purposes.

**Lemma 4.** *Assume the setting of Lemma 3 and retain the notation therein. Then, for all $\delta \in (0,1)$,*

$$\mathbb{P}\left(S_n \le 2\mathbb{E}(S_n) + \sqrt{2n\sigma^2 \log(1/\delta)} + \frac{4U}{3}\log(1/\delta)\right) \ge 1 - \delta,$$

*where $S_n$ can be either equal to $\sup_{f\in\mathcal{F}} \sum_{i=1}^n f(Z_i)$ or $\sup_{f\in\mathcal{F}} |\sum_{i=1}^n f(Z_i)|$.*

*Proof.* We take $t = \log(1/\delta)$ and bound the square-root term in (5) as follows:

$$\sqrt{2(n\sigma^2 + 2U\mathbb{E}(S_n))\log(1/\delta)} \le \sqrt{2n\sigma^2 \log(1/\delta)} + \sqrt{4U\,\mathbb{E}(S_n)\log(1/\delta)}$$
$$\le \sqrt{2n\sigma^2 \log(1/\delta)} + \mathbb{E}(S_n) + U\log(1/\delta).$$

Apply (5) in Lemma 3 with $t = \log(1/\delta)$ to complete the proof. $\square$

## A.2   Inequalities for the convolution of density functions

The first lemma is the key ingredient in order to control variance terms in integrated $L^1$−deviations of kernel density estimators. It holds under simple conditions on the tails of $f$ and $K$; in particular, it holds if $f$ and $K$ have compact support, and it holds under either Assumptions 4-5 or under Assumptions 6-7. Recall the notation $M_q(g) := \int \|z\|_2^q g(z)dz$.

**Lemma 5.** *Let $f, K : \mathbb{R}^d \to \mathbb{R}$ be two density functions such that $M_{d+\varepsilon}(f + K) < \infty$ for some $\varepsilon > 0$. Then it holds that*

$$\int \sqrt{f * K_h(x)}\,dx \le \sqrt{V_d}\left(1 + \sqrt{\frac{d}{\varepsilon}2^{d+\varepsilon-1}(M_{d+\varepsilon}(f) + h^{d+\varepsilon}M_{d+\varepsilon}(K))}\right)$$
$$\le c(1 + h^{(d+\varepsilon)/2})$$

*where $c = C_{d,\varepsilon}(1 + \sqrt{M_{d+\varepsilon}(f + K)})$ and $C_{d,\varepsilon}$ is a constant that depends on $d$ and $\varepsilon$ only.*

With this lemma at our disposal and the notation of our paper, we define $c_{y,d,\varepsilon} = C_{d,\varepsilon}(1 + \sqrt{M_{d+\varepsilon}(f_y + K)})$, which is a constant that appears in Theorem 6 and that depends on $d$, $\varepsilon$, $f_y$ and $K$ only. Extensions of this lemma may be found in Holmström and Klemelä [1992, Lemma 7 and Proposition 8], where an $L^1$-rate of convergence is established for the kernel density estimator; the above version is sufficient for our purposes.

*Proof.* Let $g$ be a density function on $\mathbb{R}^d$. By the Cauchy-Schwarz inequality, it holds

$$\int_{\|x\|_2>1} \sqrt{g(x)}\,dx \le \sqrt{\int_{\|x\|_2>1} \|x\|_2^{d+\varepsilon}g(x)\,dx}\sqrt{\int_{\|x\|_2>1} \|x\|_2^{-(d+\varepsilon)}\,dx}.$$

The second integral in the right-hand side can be calculated using polar coordinates:

$$\int_{\|x\|_2>1} \|x\|_2^{-(d+\varepsilon)}\,dx = dV_d \int_{\rho>1} \rho^{-(d+\varepsilon)}\rho^{d-1}\,d\rho = \frac{dV_d}{\varepsilon}$$

where $V_d$ is the volume of the unit ball in $\mathbb{R}^d$. Besides, by the Jensen inequality,

$$\int_{\|x\|_2\le1} \sqrt{g(x)}\frac{dx}{V_d} \le \sqrt{\int_{\|x\|_2\le1} g(x)\frac{dx}{V_d}} \le \sqrt{\frac{1}{V_d}}.$$

As a consequence

$$\int \sqrt{g(x)}\,dx = \int_{\|x\|_2>1} \sqrt{g(x)}\,dx + \int_{\|x\|_2\le1} \sqrt{g(x)}\,dx \le \sqrt{V_d}\left(1 + \sqrt{\frac{d}{\varepsilon}\int \|x\|_2^{d+\varepsilon}g(x)\,dx}\right)$$

It suffices to apply the above inequality to the density $g : x \mapsto f * K_h(x) = \int f(x-z)K_h(z)dz$ and to notice that

$$
\begin{aligned}
\int \|x\|_2^{d+\varepsilon} f * K_h(x)dx &= \iint \|x\|_2^{d+\varepsilon} f(x-z)K_h(z)dx\,dz \\
&= \iint \|s+z\|_2^{d+\varepsilon} f(s)K_h(z)ds\,dz \\
&\leq 2^{d+\varepsilon-1}\left(\iint \|s\|_2^{d+\varepsilon} f(s)K_h(z)ds\,dz + \iint \|z\|_2^{d+\varepsilon} f(s)K_h(z)ds\,dz\right) \\
&\leq 2^{d+\varepsilon-1}\left(M_{d+\varepsilon}(f) + h^{d+\varepsilon}M_{d+\varepsilon}(K)\right)
\end{aligned}
$$

to complete the proof. $\square$

We turn to two lemmas dedicated to the control of bias terms in integrated $L^1-$deviations of kernel density estimators. These lemmas are in the same spirit as Lemma 3 in Giné and Nickl [2008] as well as Lemma 6 in Delyon and Portier [2016], where similar quantities (these articles work with respect to a probability measure instead of the Lebesgue measure) are analyzed using high-order kernels to benefit from the smoothness of $f$. Our approach is somewhat different as we rely on standard kernels under twice differentiability at most. The rate of convergence obtained is therefore slower but the results have wider scope. Note also that Proposition 3 and Corollary 1 in Giné and Nickl [2008] provide (without rate) the convergence to 0 of similar quantities. One of the two statements below, which shall be used under Assumptions 4 and 5, is taken from Holmström and Klemelä [1992, Proposition 4].

**Lemma 6** (Proposition 4 in Holmström and Klemelä [1992])**.** *Suppose that $K$ is a symmetric density function on $\mathbb{R}^d$, that $M_2(K) < \infty$ and $f \in W^{2,1}(\mathbb{R}^d)$. Then*

$$
\int |f * K_h(x) - f(x)|dx \leq h^2 \phi
$$

*where*

$$
\phi = \frac{1}{2}\left(\sum_{i=1}^d \int |\partial_{ii}^2 f(x)|dx \int x_i^2 K(x)dx + 2\sum_{1\leq i<j\leq d} \int |\partial_{ij}^2 f(x)|dx \int |x_i x_j| K(x)dx\right)
$$

*is a finite constant that depends on the $L^1$-norm of the second weak partial derivatives $\partial_{ij}^2 f$ ($1 \leq i,j \leq d$) of $f$ and on $K$ only.*

With this lemma at our disposal and the notation of our paper, we define $\phi_y$ as being the constant $\phi$ appearing in (ii) with $f = f_y$, which is a constant that appears in Theorem 6 and that depends on $f_y$ and $K$ only.

Our next and final lemma is analogous to Lemma 6 but only requires the existence and integrability of the weak gradient of $f$, so that it can be applied under Assumptions 6 and 7. It should be clear that this result is not a straightforward consequence of the results of Holmström and Klemelä [1992], because it is not assumed below that $f$ is smooth on the whole of $\mathbb{R}^d$.

**Lemma 7.** *Suppose that*

- *$K$ is a symmetric density function supported on $B(0,1)$.*

- *$f$ has a support $S$ such that there are $\kappa, r_0 > 0$ with $\lambda(\partial S + B(0,r)) \leq \kappa r$ for all $r \in (0, r_0)$.*

- *$f \in W^{1,1}(U)$, where $U$ denotes the interior of $S$.*

- *$f$ is bounded on $\partial S + B(0, 2r_0)$.*

*If $h \leq r_0$, then it holds that*

$$
\int |f * K_h(x) - f(x)|dx \leq h\psi,
$$

*where, letting $\nabla f$ denote the weak gradient of $f$,*

$$
\psi = M_1(K)\int_S \|\nabla f(x)\|_2 dx + 2\kappa \sup_{z\in \partial S + B(0,2r_0)} f(z).
$$

With this lemma at our disposal and the notation of our paper, we define $\psi_y$ as being the constant $\psi$ with $f = f_y$, which is a constant that appears in Theorem 6 and that depends on $f_y$ and $K$ only.

*Proof.* Fix $h \leq r_0$. Write

$$f * K_h(x) - f(x) = \int (f(x - y) - f(x))K_h(y)dy = \int (f(x + y) - f(x))K_h(y)dy$$

by the symmetry assumption on $K$. Let $T = \partial S + B(0, h)$. Since $K$ is supported on $B(0, 1)$, for $f * K_h(x) - f(x)$ to be nonzero it is necessary that either $x \in S \cap T^c$ or $x \in T$. Moreover, the assumption on $S$ ensures that $\partial S = S \setminus U$ has Lebesgue measure zero. Hence we have

$$\int |f * K_h(x) - f(x)|dx \leq \int_{U \cap T^c} \left| \int_{B(0,h)} (f(x + y) - f(x))K_h(y)dy \right| dx$$

$$+ \int_T \left| \int_{B(0,h)} (f(x + y) - f(y))K_h(y)dy \right| dx. \quad (6)$$

We start by dealing with the first integral in the right-hand side of (6). By the Meyers-Serrin theorem [Meyers and Serrin, 1964], there exists a sequence $(f_n)$ of functions that are infinitely differentiable on $U$ and such that $f_n \to f$ in $W^{1,1}(U)$, that is, $\int_U (|f_n(x) - f(x)| + \|\nabla f_n(x) - \nabla f(x)\|_2)dx \to 0$. Note that when $x \in U \cap T^c$ and $y \in B(0, h)$, one has $x + y \in U$, and then obviously

$$\int_{U \cap T^c} \left| \int_{B(0,h)} (f_n(x + y) - f_n(x))K_h(y)dy \right| dx$$

$$= \int_{U \cap T^c} \left| \int_{B(0,h)} \int_0^1 y^\top \nabla f_n(x + ty)K_h(y)dt\, dy \right| dx.$$

This leads, for any $n$, to the inequality

$$\int_{U \cap T^c} \left| \int_{B(0,h)} (f(x + y) - f(x))K_h(y)dy \right| dx$$

$$\leq \int_{U \cap T^c} \left| \int_{B(0,h)} \int_0^1 y^\top \nabla f(x + ty)K_h(y)dt\, dy \right| dx$$

$$+ \int_{U \cap T^c} \left| \int_{B(0,h)} ((f_n - f)(x + y) - (f_n - f)(x))K_h(y)dy \right| dx$$

$$+ \int_{U \cap T^c} \left| \int_{B(0,h)} \int_0^1 y^\top \nabla (f_n - f)(x + ty)K_h(y)dt\, dy \right| dx. \quad (7)$$

Clearly

$$\int_{U \cap T^c} \left| \int_{B(0,h)} ((f_n - f)(x + y) - (f_n - f)(x))K_h(y)dy \right| dx \leq 2 \int_U |f_n(x) - f(x)|dx$$

and

$$\int_{U \cap T^c} \left| \int_{B(0,h)} \int_0^1 y^\top \nabla (f_n - f)(x + ty)K_h(y)dt\, dy \right| dx \leq h M_1(K) \int_U \|\nabla f_n(x) - \nabla f(x)\|_2 dx.$$

Both of these upper bounds converge to 0 as $n \to \infty$. Then, letting $n \to \infty$ in (7), we get

$$\int_{U \cap T^c} \left| \int_{B(0,h)} (f(x + y) - f(x))K_h(y)dy \right| dx$$

$$\leq \int_{U \cap T^c} \left| \int_{B(0,h)} \int_0^1 y^\top \nabla f(x + ty)K_h(y)dt\, dy \right| dx.$$

Now by a change of variables, symmetry of $K$, and the Cauchy-Schwarz inequality,

$$\left| \int_{B(0,h)} \int_0^1 y^\top \nabla f(x + ty) K_h(y) dt \, dy \right|$$

$$= \left| \int_{B(0,th)} \nabla f(x - u)^\top \left( \int_0^1 t^{-1-d} K_h(u/t) dt \right) u du \right|$$

$$\leq \int_{B(0,th)} \|\nabla f(x - u)\|_2 \left( \int_0^1 t^{-1-d} K_h(u/t) dt \right) \|u\|_2 du$$

$$\leq (\|\nabla f\|_2 * \|L\|_2)(x)$$

where $L(u) = (\int_0^1 t^{-1-d} K_h(u/t) dt) u$. Note that $\int \|L(u)\|_2 du = \int \|v\|_2 K_h(v) dv = h M_1(K)$. As a consequence

$$\int_{U \cap T^c} \left| \int_{B(0,h)} (f(x + y) - f(x)) K_h(y) dy \right| dx \leq \int_{U \cap T^c} (\|\nabla f\|_2 * \|L\|_2)(x) dx$$

$$\leq \int_S \|\nabla f(x)\|_2 dx \int \|L(u)\|_2 du$$

$$= h M_1(K) \int_S \|\nabla f(x)\|_2 dx. \qquad (8)$$

Concerning the second integral in (6), since there $x \in T$ and $y \in B(0, h)$, it holds that the distance of $x + y$ to $\partial S$ is at most $2h$ and then $|f(x + y) - f(x)| \leq 2 \sup_{z \in \partial S + B(0,2h)} f(z)$. Plugging this along with (8) into (6), we find, for $h \leq r_0$, that

$$\int |f * K_h(x) - f(x)| dx \leq h M_1(K) \int_S \|\nabla f(x)\|_2 dx + 2\lambda(T) \sup_{z \in \partial S + B(0,2r_0)} f(z).$$

The assumption on $S$ yields $\lambda(T) = \lambda(\partial S + B(0, h)) \leq \kappa h$ and then

$$\int |f * K_h(x) - f(x)| dx \leq h \left( M_1(K) \int_S \|\nabla f(x)\|_2 dx + 2\kappa \sup_{z \in \partial S + B(0,2r_0)} f(z) \right)$$

as announced. □

## B  Proof of Theorem 1

Let

$$Z^*(\mathcal{F}) = Z^*_{\text{Smote}}(\mathcal{F}) := \sup_{G \in \mathcal{F}} |\mu^*_{\text{Smote}}(G) - \mathbb{E}_1[G(X)]|.$$

Given that $n_1 > 0$, we decompose the supremum as $Z^*(\mathcal{F}) \leq Z^*_1(\mathcal{F}) + Z^*_2(\mathcal{F})$, with

$$Z^*_1(\mathcal{F}) = \sup_{G \in \mathcal{F}} \left| \frac{1}{m} \sum_{i=1}^m G(X^*_{1i}) - \frac{1}{n_1} \sum_{i=1}^{n_1} G(X_{1i}) \right|$$

$$\text{and } Z^*_2(\mathcal{F}) = \sup_{G \in \mathcal{F}} \left| \frac{1}{n_1} \sum_{i=1}^{n_1} G(X_{1i}) - \mathbb{E}_1[G(X)] \right|.$$

We further bound each term on the right-hand side separately, beginning with the first one. Let us recall

$$\hat{\sigma}_1^2(\mathcal{F}) = \sup_{G \in \mathcal{F}} \left[ \frac{1}{n_1} \sum_{i=1}^{n_1} (G(X_{1i}))^2 - \left( \frac{1}{n_1} \sum_{i=1}^{n_1} G(X_{1i}) \right)^2 \right].$$

**Lemma 8.** *Let $\mathcal{D}_n = \{(X_i, Y_i)\}_{i=1}^n$ be a set of $n$ i.i.d. samples drawn from a distribution $\mathbb{P}$, and let $\{X^*_{1i}\}_{i=1}^m$ be $m$ i.i.d. samples generated according to the* SMOTE *algorithm* (1)*. Suppose that*

the minority class distribution $\mathbb{P}_1$ satisfies Assumption 1. Furthermore, let $\mathcal{F}$ be a function class satisfying Assumptions 2 and 3. Fix $\delta \in (0, 1/3)$. Then, on the event $\{n_1 > 0\}$, we have

$$\mathbb{P}\left( Z_1^*(\mathcal{F}) \leq 4\mathcal{R}_m(\mathcal{F}) + \sqrt{\frac{2\hat{\sigma}_1^2(\mathcal{F})\log(1/\delta)}{m}} + \frac{8B}{3m}\log(1/\delta) + L\left(\frac{6}{C_d}\right)^{1/d}\left(\frac{k_\delta}{n_1}\right)^{1/d} \middle| Y_{1:n} \right)$$
$$\geq 1 - 3\delta.$$

*Proof.* First, recall that when $n_1 > 0$ then at each iteration $i$, $X_{1i}^*$ is drawn uniformly at random on the line linking $\tilde{X}_{1i}$ to $\overline{X}_{1i}$, where $\tilde{X}_{1i}$ is drawn uniformly at random from the minority class samples $\{X_{1i}\}_{1 \leq i \leq n_1}$. On the event $\{n_1 > 0\}$, we then consider the following decomposition:

$$Z_1^*(\mathcal{F}) \leq \sup_{G \in \mathcal{F}}\left| \frac{1}{m}\sum_{i=1}^m \left\{ G(X_{1i}^*) - G(\tilde{X}_{1i}) \right\} \right| + \sup_{G \in \mathcal{F}}\left| \frac{1}{m}\sum_{i=1}^m G(\tilde{X}_{1i}) - \frac{1}{n_1}\sum_{i=1}^{n_1} G(X_{1i}) \right|. \quad (9)$$

As for the first term in (9), using the Lipschitz property of the function class $\mathcal{F}$ stated in Assumption 2, we have

$$\sup_{G \in \mathcal{F}}\left| \frac{1}{m}\sum_{i=1}^m \left\{ G(X_{1i}^*) - G(\tilde{X}_{1i}) \right\} \right| \leq \frac{1}{m}\sum_{i=1}^m \sup_{G \in \mathcal{F}}\left| G(X_{1i}^*) - G(\tilde{X}_{1i}) \right| \leq \frac{1}{m}\sum_{i=1}^m L\|X_{1i}^* - \tilde{X}_{1i}\|_2.$$

According to the SMOTE procedure described in Section 2.2, we have $\|X_{1i}^* - \tilde{X}_{1i}\|_2 \leq r_k^{(i)}(\tilde{X}_{1i})$ where $r_k^{(i)}(\tilde{X}_{1i})$ denotes the distance to the $k$-th nearest neighbor of $\tilde{X}_{1i}$ among the sample $S^{(i)} = \{X_{11}, \ldots, X_{1n_1}\}\backslash\{\tilde{X}_{1i}\}$. Note that, by construction, $r_k^{(i)}(\tilde{X}_{1i}) = r_{k+1}(\tilde{X}_{1i})$, with $r_{k+1}$ introduced in Lemma 1 and representing the distance to the $(k+1)$-th nearest neighbor within the full sample $\{X_{11}, \ldots, X_{1n_1}\}$. It follows, when $n_1 > 1$, that

$$\sup_{G \in \mathcal{F}}\left| \frac{1}{m}\sum_{i=1}^m \left\{ G(X_{1i}^*) - G(\tilde{X}_{1i}) \right\} \right| \leq L \max_{1 \leq i \leq m} r_k^{(i)}(\tilde{X}_{1i}) \leq L \sup_{x \in \text{supp}(\mathbb{P}_1)} r_{k+1}(x). \quad (10)$$

Conditionally on $Y_{1:n}$ and given that $n_1 > 1$, $\{X_{11}, \ldots, X_{1n_1}\}$ is an i.i.d. sample of $n_1$ elements with common distribution $\mathbb{P}_1$. Then applying Lemma 1, we obtain that, whenever $n_1 > 1$,

$$\mathbb{P}\left( \sup_{x \in \text{supp}(\mathbb{P}_1)} r_{k+1}(x) > \left(\frac{3}{C_d}\right)^{1/d}\left(\frac{\max(k+1, (d+1)\log(2n_1) + \log(8/\delta))}{n_1}\right)^{1/d} \middle| Y_{1:n} \right) \leq \delta.$$

Clearly $\max(k+1, (d+1)\log(2n_1) + \log(8/\delta)) \leq \max(2k, (d+1)\log(2n_1) + \log(8/\delta)) \leq 2k_\delta$ so that, when $n_1 > 1$,

$$\mathbb{P}\left( \sup_{x \in \text{supp}(\mathbb{P}_1)} r_{k+1}(x) > \left(\frac{6}{C_d}\right)^{1/d}\left(\frac{k_\delta}{n_1}\right)^{1/d} \middle| Y_{1:n} \right) \leq \delta.$$

Dealing with the case $n_1 = 1$ separately, for which the inequality below is trivial, and using (10), we obtain that whenever $n_1 > 0$,

$$\mathbb{P}\left( \sup_{G \in \mathcal{F}}\left| \frac{1}{m}\sum_{i=1}^m \left\{ G(X_{1i}^*) - G(\tilde{X}_{1i}) \right\} \right| \leq L\left(\frac{6}{C_d}\right)^{1/d}\left(\frac{k_\delta}{n_1}\right)^{1/d} \middle| Y_{1:n} \right) \geq 1 - \delta.$$

For the second term in (9), we further work conditionally on $\mathcal{D}_n$ while assuming that $n_1 > 0$. For any $G \in \mathcal{F}$, define $\psi_G(x) = G(x) - \mu(G)$, where $\mu(G) = \frac{1}{n_1}\sum_{i=1}^{n_1} G(X_{1i})$. Note that since $\tilde{X}_{11}$ is drawn uniformly at random from $\{X_{1i}\}_{1 \leq i \leq n_1}$, we have $\mathbb{E}(G(\tilde{X}_{11}) \mid \mathcal{D}_n) = \mu(G)$, so $\psi_G(\tilde{X}_{11})$ is centered given $\mathcal{D}_n$. Let

$$S_m = \sup_{G \in \mathcal{F}} \frac{1}{m}\sum_{i=1}^m \psi_G(\tilde{X}_{1i}) = \sup_{G \in \mathcal{F}}\left( \frac{1}{m}\sum_{i=1}^m G(\tilde{X}_{1i}) - \frac{1}{n_1}\sum_{i=1}^{n_1} G(X_{1i}) \right).$$

The class of functions $\{\psi_G, G \in \mathcal{F}\}$ satisfies the assumptions of Lemma 3 (with the centering assumption understood conditionally on $\mathcal{D}_n$ and $n_1 > 0$), and

$$\sup_{G \in \mathcal{F}} \mathrm{Var}\big(\psi_G(\tilde{X}_{11}) \mid \mathcal{D}_n\big) = \sup_{G \in \mathcal{F}} \mathbb{E}\left(\big(G(\tilde{X}_{11}) - \mu(G)\big)^2\right)$$

$$= \sup_{G \in \mathcal{F}} \left[\frac{1}{n_1} \sum_{i=1}^{n_1}(G(X_{1i}))^2 - \left(\frac{1}{n_1}\sum_{i=1}^{n_1} G(X_{1i})\right)^2\right] = \hat{\sigma}_1^2(\mathcal{F}).$$

Using the symmetrization technique, we derive a bound for $\mathbb{E}[S_m|\mathcal{D}_n]$ in order to apply Lemma 4. Let $\{\tilde{X}'_{1i}\}_{i=1}^m$ be an independent copy of $\{\tilde{X}_{1i}\}_{i=1}^m$, given $\mathcal{D}_n$, and let $\{\varepsilon_i\}_{i=1}^m$ be independent Rademacher random variables that are independent of the $\{\tilde{X}'_{1i}\}_{i=1}^m$, $\{\tilde{X}_{1i}\}_{i=1}^m$, and of $\mathcal{D}_n$. Then:

$$\mathbb{E}[S_m \mid \mathcal{D}_n] \le \mathbb{E}\left[\sup_{G \in \mathcal{F}} \frac{1}{m}\sum_{i=1}^m \big(G(\tilde{X}_{1i}) - G(\tilde{X}'_{1i})\big)\,\Big|\, \mathcal{D}_n\right] \qquad \text{(ghost sample)}$$

$$= \mathbb{E}\left[\sup_{G \in \mathcal{F}} \frac{1}{m}\sum_{i=1}^m \varepsilon_i \big(G(\tilde{X}_{1i}) - G(\tilde{X}'_{1i})\big)\,\Big|\, \mathcal{D}_n\right] \qquad \text{(symmetrization)}$$

$$\le 2\,\mathbb{E}\left[\sup_{G \in \mathcal{F}} \frac{1}{m}\sum_{i=1}^m \varepsilon_i\, G(\tilde{X}_{1i})\,\Big|\, \mathcal{D}_n\right]$$

$$\le 2\,\mathcal{R}_m(\mathcal{F}),$$

where the last inequality follows from Assumption 3. Then, by Lemma 4 applied to $S_m$, we obtain

$$\mathbb{P}\left(\sup_{G \in \mathcal{F}}\left(\frac{1}{m}\sum_{i=1}^m G(\tilde{X}_{1i}) - \frac{1}{n_1}\sum_{i=1}^{n_1} G(X_{1i})\right) \le 4\mathcal{R}_m(\mathcal{F}) + \sqrt{\frac{2\hat{\sigma}_1^2(\mathcal{F})\log(1/\delta)}{m}}\right.$$

$$\left. + \frac{8B}{3m}\log(1/\delta)\,\Big|\, \mathcal{D}_n\right) \ge 1 - \delta.$$

By combining this with the same argument applied to $-\psi_G = \psi_{-G}$, and using the symmetry of the $\varepsilon_i$, we obtain

$$\mathbb{P}\left(\sup_{G \in \mathcal{F}}\left|\frac{1}{m}\sum_{i=1}^m G(\tilde{X}_{1i}) - \frac{1}{n_1}\sum_{i=1}^{n_1} G(X_{1i})\right| \le 4\mathcal{R}_m(\mathcal{F}) + \sqrt{\frac{2\hat{\sigma}_1^2(\mathcal{F})\log(1/\delta)}{m}}\right.$$

$$\left. + \frac{8B}{3m}\log(1/\delta)\,\Big|\, \mathcal{D}_n\right) \ge 1 - 2\delta,$$

which implies that when $n_1 > 0$, we have

$$\mathbb{P}\left(\sup_{G \in \mathcal{F}}\left|\frac{1}{m}\sum_{i=1}^m G(\tilde{X}_{1i}) - \frac{1}{n_1}\sum_{i=1}^{n_1} G(X_{1i})\right| \le 4\mathcal{R}_m(\mathcal{F}) + \sqrt{\frac{2\hat{\sigma}_1^2(\mathcal{F})\log(1/\delta)}{m}}\right.$$

$$\left. + \frac{8B}{3m}\log(1/\delta)\,\Big|\, Y_{1:n}\right) \ge 1 - 2\delta$$

by integrating out the conditional probability with respect to $X_1, \ldots, X_n$. The proof is complete. $\square$

**Lemma 9.** *Let $\mathcal{D}_n = \{(X_i, Y_i)\}_{i=1}^n$ be $n$ i.i.d. samples drawn from a distribution $\mathbb{P}$, and let $\mathcal{F}$ be a function class satisfying Assumptions 2 and 3. Fix $\delta \in (0, 1/2)$. Then, whenever $n_1 > 0$, and if $\sigma_1^2(\mathcal{F}) = \sup_{G \in \mathcal{F}} \mathrm{Var}[G(X)|Y = 1]$, we have*

$$\mathbb{P}\left(Z_2^*(\mathcal{F}) \le 4\mathcal{R}_{n_1}(\mathcal{F}) + \sqrt{\frac{2\sigma_1^2(\mathcal{F})\log(1/\delta)}{n_1}} + \frac{8B}{3n_1}\log(1/\delta)\,\Big|\, Y_{1:n}\right) \ge 1 - 2\delta.$$

*Proof.* Conditioning on the label sequence $Y_{1:n} = (Y_1, \ldots, Y_n)$ makes $n_1$ deterministic while leaving the $\{X_{1i}\}_{1 \le i \le n_1}$ i.i.d.. Following the second part of the proof of Lemma 8, define $\psi_G(x) = G(x) - \mathbb{E}_1[G(X)]$ and, when $n_1 > 0$,

$$S_{n_1} = \sup_{G \in \mathcal{F}} \frac{1}{n_1} \sum_{i=1}^{n_1} \psi_G(X_{1i}).$$

Then, on $\{n_1 > 0\}$, we obtain by the same argument that $\mathbb{E}[S_{n_1}|Y_{1:n}] \leq 2\mathcal{R}_{n_1}(\mathcal{F})$ and, by Lemma 4, we get

$$\mathbb{P}\left(S_{n_1} \leq 4\mathcal{R}_{n_1}(\mathcal{F}) + \sqrt{\frac{2\sigma_1^2(\mathcal{F})\log(1/\delta)}{n_1}} + \frac{8B}{3n_1}\log(1/\delta) \,\middle|\, Y_{1:n}\right) \geq 1 - \delta.$$

Repeating the argument for $-\psi_G$, we obtain the bound

$$\mathbb{P}\left(\sup_{G \in \mathcal{F}}\left|\frac{1}{n_1}\sum_{i=1}^{n_1} G(X_{1i}) - \mathbb{E}_1[G(X)]\right| \leq 4\mathcal{R}_{n_1}(\mathcal{F}) + \sqrt{\frac{2\sigma_1^2(\mathcal{F})\log(1/\delta)}{n_1}}\right.$$
$$\left. + \frac{8B}{3n_1}\log(1/\delta) \,\middle|\, Y_{1:n}\right) \geq 1 - 2\delta$$

as required. $\qquad\square$

*End of the proof of Theorem 1.* Let $t > 0$. We have

$$\mathbb{P}\left(\{Z^*(\mathcal{F}) > t\}\,|\,Y_{1:n}\right) = \mathbb{P}\left(\{Z^*(\mathcal{F}) > t\}\,|\,Y_{1:n}\right)1_{\{n_1>0\}} + \mathbb{P}\left(\{Z^*(\mathcal{F}) > t\}\,|\,Y_{1:n}\right)1_{\{n_1=0\}}.$$

However, from $Z^*(\mathcal{F}) \leq Z_1^*(\mathcal{F}) + Z_2^*(\mathcal{F})$, the union bound ensures that, if $t_1 + t_2 = t$,

$$\mathbb{P}\left(\{Z^*(\mathcal{F}) > t\}\,|\,Y_{1:n}\right)1_{\{n_1>0\}}$$
$$\leq \mathbb{P}\left(\{Z_1^*(\mathcal{F}) > t_1\}\,|\,Y_{1:n}\right)1_{\{n_1>0\}} + \mathbb{P}\left(\{Z_2^*(\mathcal{F}) > t_2\}\,|\,Y_{1:n}\right)1_{\{n_1>0\}}.$$

By Lemmas 8 and 9, it follows that, whenever $n_1 > 0$,

$$\mathbb{P}\left(\{Z^*(\mathcal{F}) > t\}\,|\,Y_{1:n}\right) \leq 5\delta, \tag{11}$$

where $t_1$ and $t_2$ are respectively chosen as the upper bounds in Lemmas 8 and 9, resulting in

$$t = 4\mathcal{R}_m(\mathcal{F}) + 4\mathcal{R}_{n_1}(\mathcal{F}) + L\left(\frac{6}{C_d}\right)^{1/d}\left(\frac{k_\delta}{n_1}\right)^{1/d}$$
$$+ \sqrt{\frac{2\hat{\sigma}_1^2(\mathcal{F})\log(1/\delta)}{m}} + \sqrt{\frac{2\sigma_1^2(\mathcal{F})\log(1/\delta)}{n_1}} + \left(\frac{8B}{3m} + \frac{8B}{3n_1}\right)\log(1/\delta).$$

When $n_1 = 0$, we set $t = +\infty$ and thus $P\left(\{Z^*(\mathcal{F}) > t\}\,|\,Y_{1:n}\right) = 0$. As a consequence, we obtain $\mathbb{P}(Z^*(\mathcal{F}) > t) = \mathbb{E}(\mathbb{P}\left(\{Z^*(\mathcal{F}) > t\}\,|\,Y_{1:n}\right)1_{\{n_1>0\}}) \leq 5\delta\mathbb{P}(n_1 > 0) \leq 5\delta$, which completes the proof. $\qquad\square$

## C  Proof of Theorem 3

We start exactly as in the proof of Theorem 1. On the event $n_1 > 0$, let

$$Z^*(\mathcal{F}) = Z_{\text{KDEO}}^*(\mathcal{F}) := \sup_{G \in \mathcal{F}}|\mu_{\text{KDEO}}^*(G) - \mathbb{E}_1[G(X)]|$$

and write $Z^*(\mathcal{F}) \leq Z_1^*(\mathcal{F}) + Z_2^*(\mathcal{F})$, with

$$Z_1^*(\mathcal{F}) = \sup_{G \in \mathcal{F}}\left|\frac{1}{m}\sum_{i=1}^{m} G(X_{1i}^*) - \frac{1}{n_1}\sum_{i=1}^{n_1} G(X_{1i})\right|$$

$$\text{and } Z_2^*(\mathcal{F}) = \sup_{G \in \mathcal{F}}\left|\frac{1}{n_1}\sum_{i=1}^{n_1} G(X_{1i}) - \mathbb{E}_1[G(X)]\right|.$$

The second term has already been controlled in Lemma 9. The first term is the focus of the next lemma.

**Lemma 10.** *Let $\mathcal{D}_n = \{(X_i, Y_i)\}_{i=1}^n$ be a set of $n$ i.i.d. samples drawn from a distribution $\mathbb{P}$, and let $\{X_{1i}^*\}_{i=1}^m$ be $m$ i.i.d. samples generated according to KDE-based oversampling (2). Let $\mathcal{F}$ be a function class satisfying Assumptions 2 and 3. Fix $\delta \in (0, 1/3)$. Then, on the event $\{n_1 > 0\}$, we have*

$$\mathbb{P}\left(Z_1^*(\mathcal{F}) \leq 4\mathcal{R}_m(\mathcal{F}) + \sqrt{\frac{2\hat{\sigma}_1^2(\mathcal{F})\log(1/\delta)}{m}} + 5LhM_1(K) + \frac{9B}{m}\log(1/\delta) \,\middle|\, Y_{1:n}\right) \geq 1 - 3\delta.$$

*Proof.* Analogously to the proof of Lemma 8, we work on the event $\{n_1 > 0\}$ to write, as in (9):

$$Z_1^*(\mathcal{F}) \leq \sup_{G \in \mathcal{F}} \left| \frac{1}{m} \sum_{i=1}^m \left\{ G(X_{1i}^*) - G(\tilde{X}_{1i}) \right\} \right| + \sup_{G \in \mathcal{F}} \left| \frac{1}{m} \sum_{i=1}^m G(\tilde{X}_{1i}) - \frac{1}{n_1} \sum_{i=1}^{n_1} G(X_{1i}) \right|.$$

For the second term in the right-hand side, we obtained in the proof of Lemma 8 that

$$\mathbb{P}\left( \sup_{G \in \mathcal{F}} \left| \frac{1}{m} \sum_{i=1}^m G(\tilde{X}_{1i}) - \frac{1}{n_1} \sum_{i=1}^{n_1} G(X_{1i}) \right| \leq 4 \mathcal{R}_m(\mathcal{F}) + \sqrt{\frac{2 \hat{\sigma}_1^2(\mathcal{F}) \log(1/\delta)}{m}} \right.$$
$$\left. + \frac{8B}{3m} \log(1/\delta) \;\Big|\; Y_{1:n} \right) \geq 1 - 2\delta \tag{12}$$

on the event $\{n_1 > 0\}$. The first term needs a particular treatment because, by contrast with SMOTE, the KDE-based perturbation $hW_i$ used to generate $X_{1i}^*$ from $\tilde{X}_{1i}$ is not bounded anymore. We further decompose this first term. Note that conditionally on the initial sample $\mathcal{D}_n$ and $\{n_1 > 0\}$, the $\Delta_i^* = \Delta_i^*(G) = G(X_{1i}^*) - G(\tilde{X}_{1i})$ are i.i.d. and that, for any $G \in \mathcal{F}$,

$$\mathbb{E}[|\Delta_1^*||\mathcal{D}_n] = \frac{1}{n_1} \sum_{i=1}^{n_1} \int |G(X_{1i} + hw) - G(X_{1i})| K(w) dw \leq Lh \int \|w\|_2 K(w) dw = LhM_1(K). \tag{13}$$

We then have, when $n_1 > 0$,

$$\sup_{G \in \mathcal{F}} \left| \frac{1}{m} \sum_{i=1}^m (G(X_{1i}^*) - G(\tilde{X}_{1i})) \right| \leq \sup_{G \in \mathcal{F}} \left| \frac{1}{m} \sum_{i=1}^m (\Delta_i^* - \mathbb{E}[\Delta_i^*|\mathcal{D}_n]) \right| + LhM_1(K).$$

Since $|\Delta_i^*| \leq 2B$, we have $|\Delta_i^* - \mathbb{E}[\Delta_i^*|\mathcal{D}_n]| \leq 4B$ and, due to the above bound on $\mathbb{E}[|\Delta_1^*||\mathcal{D}_n]$, it holds that

$$\text{Var}(\Delta_1^*|\mathcal{D}_n) \leq \mathbb{E}[\Delta_1^{*2}|\mathcal{D}_n] \leq 2B\mathbb{E}[|\Delta_1^*||\mathcal{D}_n] \leq 2BLhM_1(K).$$

Applying Lemma 4 (with absolute value), we find that when $n_1 > 0$,

$$\mathbb{P}\left( \sup_{G \in \mathcal{F}} \left| \frac{1}{m} \sum_{i=1}^m (\Delta_i^* - \mathbb{E}[\Delta_i^* \mid \mathcal{D}_n]) \right| \leq 2\, \mathbb{E}\left[ \sup_{G \in \mathcal{F}} \left| \frac{1}{m} \sum_{i=1}^m (\Delta_i^* - \mathbb{E}[\Delta_i^* \mid \mathcal{D}_n]) \right| \;\Big|\; \mathcal{D}_n \right] \right.$$
$$\left. + \sqrt{\frac{4BLhM_1(K) \log(1/\delta)}{m}} + \frac{16B}{3m} \log(1/\delta) \;\Big|\; \mathcal{D}_n \right) \geq 1 - \delta.$$

Using (13) and the inequality $2\sqrt{ab} \leq a + b$ for any $a, b \geq 0$, we obtain that

$$\mathbb{P}\left( \sup_{G \in \mathcal{F}} \left| \frac{1}{m} \sum_{i=1}^m (G(X_{1i}^*) - G(\tilde{X}_{1i})) \right| \leq 5LhM_1(K) + \frac{19B}{3m} \log(1/\delta) \;\Big|\; \mathcal{D}_n \right) \geq 1 - \delta.$$

Then, when $n_1 > 0$,

$$\mathbb{P}\left( \sup_{G \in \mathcal{F}} \left| \frac{1}{m} \sum_{i=1}^m (G(X_{1i}^*) - G(\tilde{X}_{1i})) \right| \leq 5LhM_1(K) + \frac{19B}{3m} \log(1/\delta) \;\Big|\; Y_{1:n} \right) \geq 1 - \delta$$

by integrating out the conditional probability with respect to $X_1, \ldots, X_n$. The result follows by recalling (12). $\qquad\square$

*End of the proof of Theorem 3.* The proof is similar to that of Theorem 1. Putting together the conclusion of Lemma 9 and that of Lemma 10, we obtain that whenever $n_1 > 0$,

$$\mathbb{P}\left( \{Z_{\text{KDEO}}^*(\mathcal{F}) > t\} \mid Y_{1:n} \right) \leq 5\delta, \tag{14}$$

where

$$t = 4\mathcal{R}_{n_1}(\mathcal{F}) + 4\mathcal{R}_m(\mathcal{F}) + 5LhM_1(K)$$
$$+ \left( \sqrt{\frac{\sigma_1^2(\mathcal{F})}{n_1}} + \sqrt{\frac{\hat{\sigma}_1^2(\mathcal{F})}{m}} \right) \sqrt{2\log\left(\frac{1}{\delta}\right)} + \frac{B}{3}\left(\frac{27}{m} + \frac{8}{n_1}\right) \log\left(\frac{1}{\delta}\right).$$

The result follows because under $n_1 = 0$, it holds that $\mathbb{P}(\{Z_{\text{KDEO}}^*(\mathcal{F}) > +\infty\} \mid Y_{1:n}) = 0$. $\qquad\square$

# D  Proof of Corollaries 2 and 4

We start by proving Corollary 2 and then we will adapt the proof to obtain Corollary 4. Let

$$\hat{R}_{1/2}(g) := \frac{1}{m + n_0} \left( 1_{\{n_1 > 0\}} \sum_{i=1}^{m} \ell(g(X_{1i}^*)) + 1_{\{n_0 > 0\}} \sum_{i=1}^{n_0} \ell(-g(X_{0i})) \right).$$

The proof starts with the standard decomposition

$$R_{1/2}(\hat{g}_{\mathcal{G}}^*) - \inf_{g \in \mathcal{G}} R_{1/2}(g) \leq 2 \sup_{g \in \mathcal{G}} |\hat{R}_{1/2}(g) - R_{1/2}(g)|.$$

This follows from, first, the inequalities

$$R_{1/2}(\hat{g}_{\mathcal{G}}^*) - \inf_{g \in \mathcal{G}} R_{1/2}(g) = R_{1/2}(\hat{g}_{\mathcal{G}}^*) - \hat{R}_{1/2}(\hat{g}_{\mathcal{G}}^*) + \hat{R}_{1/2}(\hat{g}_{\mathcal{G}}^*) - \inf_{g \in \mathcal{G}} R_{1/2}(g)$$

$$\leq \sup_{g \in \mathcal{G}} |\hat{R}_{1/2}(g) - R_{1/2}(g)| + \inf_{g \in \mathcal{G}} \hat{R}_{1/2}(g) - \inf_{g \in \mathcal{G}} R_{1/2}(g)$$

and, second, from the inequalities

$$\inf_{g \in \mathcal{G}} \hat{R}_{1/2}(g) - \inf_{g \in \mathcal{G}} R_{1/2}(g) \leq \hat{R}_{1/2}(g') - \inf_{g \in \mathcal{G}} R_{1/2}(g)$$

$$= \hat{R}_{1/2}(g') - R_{1/2}(g') + R_{1/2}(g') - \inf_{g \in \mathcal{G}} R_{1/2}(g)$$

$$\leq \sup_{g \in \mathcal{G}} |\hat{R}_{1/2}(g) - R_{1/2}(g)| + R_{1/2}(g') - \inf_{g \in \mathcal{G}} R_{1/2}(g)$$

valid for any $g' \in \mathcal{G}$. Then, by definition of $\hat{R}_{1/2}(g)$ and $R_{1/2}(g)$ and using $m = n_0$, we find, when $n_0, n_1 > 0$,

$$R_{1/2}(\hat{g}_{\mathcal{G}}^*) - \inf_{g \in \mathcal{G}} R_{1/2}(g)$$

$$\leq 2 \sup_{g \in \mathcal{G}} \frac{1}{m + n_0} \left| \sum_{i=1}^{m} \{\ell(g(X_{1i}^*)) - \mathbb{E}_1[\ell(g(X))]\} + \sum_{i=1}^{n_0} \{\ell(-g(X_{0i})) - \mathbb{E}_0[\ell(-g(X))]\} \right|$$

$$\leq \sup_{g \in \mathcal{G}} \left| \frac{1}{m} \sum_{i=1}^{m} \ell(g(X_{1i}^*)) - \mathbb{E}_1[\ell(g(X))] \right| + \sup_{g \in \mathcal{G}} \left| \frac{1}{n_0} \sum_{i=1}^{n_0} \ell(-g(X_{0i})) - \mathbb{E}_0[\ell(-g(X))] \right|$$

$$= \sup_{G \in \ell(\mathcal{G})} \left| \frac{1}{m} \sum_{i=1}^{m} G(X_{1i}^*) - \mathbb{E}_1[G(X)] \right| + \sup_{G \in \ell(-\mathcal{G})} \left| \frac{1}{n_0} \sum_{i=1}^{n_0} G(X_{0i}) - \mathbb{E}_0[G(X)] \right|$$

$$\leq \sup_{G \in \mathcal{F}} \left| \frac{1}{m} \sum_{i=1}^{m} G(X_{1i}^*) - \mathbb{E}_1[G(X)] \right| + \sup_{G \in \mathcal{F}} \left| \frac{1}{n_0} \sum_{i=1}^{n_0} G(X_{0i}) - \mathbb{E}_0[G(X)] \right|$$

$$= Z^*(\mathcal{F}) + Z_0(\mathcal{F})$$

where $Z^*(\mathcal{F})$ is introduced in the proof of Theorem 1 and $Z_0(\mathcal{F})$ is the rightmost term. Regarding $Z_0(\mathcal{F})$, we can use Lemma 9 (with $\mathbb{P}_0$ instead of $\mathbb{P}_1$) to get that, when $n_0 > 0$, it holds that

$$\mathbb{P}\left( Z_0(\mathcal{F}) > t_0 \,|\, Y_{1:n} \right) \leq 2\delta,$$

with

$$t_0 = 4\mathcal{R}_{n_0}(\mathcal{F}) + \sqrt{\frac{2\sigma_0^2(\mathcal{F})\log(1/\delta)}{n_0}} + \frac{8B}{3n_0}\log(1/\delta).$$

About the first term, $Z^*(\mathcal{F})$, it is shown in (11) that, whenever $n_1 > 0$,

$$\mathbb{P}\left( Z^*(\mathcal{F}) > t \,|\, Y_{1:n} \right) \leq 5\delta,$$

with $t$ defined just below (11). The union bound gives that, whenever $n_0, n_1 > 0$,

$$\mathbb{P}\left( R_{1/2}(\hat{g}_{\mathcal{G}}^*) - \inf_{g \in \mathcal{G}} R_{1/2}(g) > t + t_0 \,|\, Y_{1:n} \right) \leq 7\delta.$$

Then, setting $t + t_0 = +\infty$ in case $n_0 n_1 = 0$, and considering the cases $n_0 n_1 > 0$ and $n_0 n_1 = 0$ separately, we find

$$\mathbb{P}\left(R_{1/2}(\hat{g}_{\mathcal{G}}^*) - \inf_{g \in \mathcal{G}} R_{1/2}(g) > t + t_0 \,|\, Y_{1:n}\right)$$

$$= \mathbb{P}\left(R_{1/2}(\hat{g}_{\mathcal{G}}^*) - \inf_{g \in \mathcal{G}} R_{1/2}(g) > t + t_0 \,|\, Y_{1:n}\right) 1_{\{n_0 n_1 > 0\}} \le 7\delta.$$

Hence taking the expectation gives

$$\mathbb{P}\left(R_{1/2}(\hat{g}_{\mathcal{G}}^*) - \inf_{g \in \mathcal{G}} R_{1/2}(g) > t + t_0\right) \le 7\delta.$$

We obtain the statement of Corollary 2 by using that $m = n_0$ when rearranging the terms in $t + t_0$.

Concerning Corollary 4, the proof proceeds in a similar fashion, up to the application of (11). At this point, we instead use (14) to complete the argument. $\qquad\square$

# E   Proof of Theorem 5

We have

$$2R_{1/2}(\hat{g}) = \mathbb{E}_1(1_{\{\hat{g}(X)\neq 1\}}) + \mathbb{E}_0(1_{\{\hat{g}(X)\neq 0\}})$$
$$= (1 - p_1)^{-1}\mathbb{E}(1_{\{\hat{g}(X)=1\}}(1 - \eta(X))) + p_1^{-1}\mathbb{E}(1_{\{\hat{g}(X)=0\}}\eta(X))$$

by taking conditional expectations with respect to $X$. Besides, $\mathbb{E}(\eta(X)) = p_1$ and then

$$2R_{1/2}(\hat{g}) = (1 - p_1)^{-1}\mathbb{E}(1_{\{\hat{g}(X)=1\}}(1 - \eta(X))) - p_1^{-1}\mathbb{E}(1_{\{\hat{g}(X)=1\}}\eta(X)) + 1$$
$$= (1 - p_1)^{-1}p_1^{-1}\mathbb{E}(1_{\{\hat{g}(X)=1\}}(p_1 - \eta(X))) + 1.$$

It follows that

$$2(R_{1/2}(\hat{g}) - R_{1/2}(g)) = (1 - p_1)^{-1}p_1^{-1}\mathbb{E}((1_{\{\hat{g}(X)=1\}} - 1_{\{g(X)=1\}})(p_1 - \eta(X))).$$

Observe that $1_{\{\hat{g}(X)=1\}} - 1_{\{g(X)=1\}} = 1_{\{\hat{g}(X)\neq g(X)\}}(1_{\{g(X)=0\}} - 1_{\{g(X)=1\}})$ and that, by definition of $g$, $g(X) = 0$ if and only if $\eta(X) \le p_1$. As such

$$(1_{\{g(X)=0\}} - 1_{\{g(X)=1\}})(p_1 - \eta(X)) = |p_1 - \eta(X)|$$

and then

$$2(R_{1/2}(\hat{g}) - R_{1/2}(g)) = (1 - p_1)^{-1}p_1^{-1}\mathbb{E}(1_{\{\hat{g}(X)\neq g(X)\}}|p_1 - \eta(X)|)$$

$$= (1 - p_1)^{-1}\int_S 1_{\{\hat{g}(x)\neq g(x)\}}|f(x) - f_1(x)|dx$$

$$= \int_S 1_{\{\hat{g}(x)\neq g(x)\}}|f_1(x) - f_0(x)|dx$$

$$\le \int_S 1_{\{\hat{g}(x)\neq g(x)\}}|f_1(x) - f_0(x) - (\hat{f}_1(x) - \hat{f}_0(x))|dx$$

where to obtain the last inequality, we have used that $\hat{g}$ and $g$ disagree if and only if $\hat{f}_1 - \hat{f}_0$ and $f_1 - f_0$ have different signs. Conclude using the triangle inequality. $\qquad\square$

# F   Proof of Theorem 6

The proof consists in applying Theorem 5 after obtaining two bounds on the $L^1-$errors of $\hat{f}_{0s}$ and $\hat{f}_{1s}^*$. This leads to proving two preliminary results which are of interest in their own right. Recall that $\hat{f}_{yh}(x) = 0$ if $n_y = 0$.

**Proposition 7** ($L^1-$bound on class-specific kernel density estimators). *Let $\delta \in (0,1)$ and $y \in \{0,1\}$. Then, on the event $\{n_y > 0\}$, we have*

$$\mathbb{P}\left( \int |\hat{f}_{yh}(x) - f_y(x)| dx \leq c_{y,d,\varepsilon}(1 + h^{(d+\varepsilon)/2})\sqrt{\frac{M_0(K^2)}{n_y h^d}} + \sqrt{\frac{2\log(1/\delta)}{n_y}} \right.$$

$$\left. + \begin{cases} \phi_y h^2 & \text{under Assumptions 4 and 5} \\ \psi_y h & \text{if } h \leq r_0 \text{ under Assumptions 6 and 7} \end{cases} \middle| Y_{1:n} \right) \geq 1 - \delta$$

*where $c_{y,d,\varepsilon}$ (resp. $\phi_y$, $\psi_y$) is defined below Lemma 5 (resp. below Lemma 6, below Lemma 7) and depends only on $(K, f_y, d, \epsilon)$ (resp. $(K, f_y)$).*

*Proof.* Denote throughout by $Y_{1:n}$ the random vector $(Y_1, \ldots, Y_n)$. We start by applying McDiarmid's inequality (Lemma 2), with respect to the conditional probability given $Y_{1:n}$, to

$$T_{Y_{1:n}}(X_1, \ldots, X_n) = \int |\hat{f}_{yh}(x) - f_y(x)| dx.$$

Note that for any $i \in \{1, \ldots, n\}$, whenever $n_y > 0$,

$$|T_{Y_{1:n}}(X_1, \ldots, X_{i-1}, X_i, X_{i+1}, \ldots, X_n) - T_{Y_{1:n}}(X_1, \ldots, X_{i-1}, X_i', X_{i+1}, \ldots, X_n)|$$

$$\leq \frac{1_{\{Y_i=y\}}}{n_y} \int |K_h(x - X_i) - K_h(x - X_i')| dx \leq 2\frac{1_{\{Y_i=y\}}}{n_y} =: C_i.$$

Remarking that $\sum_{i=1}^n C_i^2 = (4/n_y)1_{\{n_y>0\}}$, and that the assumption of Lemma 2 is trivially correct for $n_y = 0$, we obtain by the McDiarmid inequality that with probability (conditionally on $Y_{1:n}$) at least $1 - \delta$,

$$\int |\hat{f}_{yh}(x) - f_y(x)| dx \leq \mathbb{E}\left( \int |\hat{f}_{yh}(x) - f_y(x)| dx \middle| Y_{1:n} \right) + \sqrt{\frac{2\log(1/\delta)}{n_y}}1_{\{n_y>0\}}.$$

Swapping integral and conditional expectation and using the triangle and (conditional) Jensen inequalities, we get

$$\int |\hat{f}_{yh}(x) - f_y(x)| dx \leq \int \sqrt{\mathbb{E}\left[ \left( \hat{f}_{yh}(x) - \mathbb{E}(\hat{f}_{yh}(x)|Y_{1:n}) \right)^2 \middle| Y_{1:n} \right]} dx$$

$$+ \int \left| \mathbb{E}(\hat{f}_{yh}(x)|Y_{1:n}) - f_y(x) \right| dx + \sqrt{\frac{2\log(1/\delta)}{n_y}}1_{\{n_y>0\}}. \qquad (15)$$

The first two terms depend on $Y_{1:n}$ and are further investigated on the event $\{n_y > 0\}$. For the first term above, which is the variance term, we have, by independence of the random variables $(Y_1, X_1), \ldots, (Y_n, X_n)$,

$$\mathbb{E}\left[ \left( \hat{f}_{yh}(x) - \mathbb{E}(\hat{f}_{yh}(x)|Y_{1:n}) \right)^2 \middle| Y_{1:n} \right] = \frac{1}{n_y^2} \sum_{i=1}^n \text{Var}(K_h(x - X_i)|Y_i)1_{\{Y_i=y\}}$$

$$\leq \frac{1}{n_y^2} \sum_{i=1}^n \mathbb{E}(K_h^2(x - X_i)|Y_i)1_{\{Y_i=y\}}$$

$$= \frac{1}{n_y}\mathbb{E}(K_h^2(x - X)|Y = y)$$

$$= \frac{M_0(K^2)}{n_y h^d} f_y * \tilde{K}_h(x)$$

with $\tilde{K} = K^2/M_0(K^2)$. By Lemma 5 applied to $f_y$ and $\tilde{K}$,

$$\int \sqrt{\mathbb{E}\left[ \left( \hat{f}_{yh}(x) - \mathbb{E}(\hat{f}_{yh}(x)|Y_{1:n}) \right)^2 \middle| Y_{1:n} \right]} dx \leq c_{y,d,\varepsilon}(1 + h^{(d+\varepsilon)/2})\sqrt{\frac{M_0(K^2)}{n_y h^d}}. \qquad (16)$$

For the bias term, noting that

$$\mathbb{E}(\hat{f}_{yh}(x)|Y_{1:n}) = \frac{1}{n_y}\sum_{i=1}^{n}\mathbb{E}(K_h(x-X_i)|Y_i)1_{\{Y_i=y\}} = \mathbb{E}(K_h(x-X)|Y=y) = f_y * K_h(x),$$

we apply Lemma 6 to obtain that

$$\int |\mathbb{E}(\hat{f}_{yh}(x)|Y_{1:n}) - f_y(x)|dx \le \phi_y h^2 \tag{17}$$

under Assumptions 4-5, and we use Lemma 7 to find

$$\int |\mathbb{E}(\hat{f}_{yh}(x)|Y_{1:n}) - f_y(x)|dx \le \psi_y h \tag{18}$$

under Assumptions 6-7 when $h \le r_0$. Combining (15), (16), (17) and (18), we have therefore shown that the event

$$E = \left\{ \int |\hat{f}_{yh}(x) - f_y(x)|dx \le c_{y,d,\varepsilon}(1+h^{(d+\varepsilon)/2})\sqrt{\frac{M_0(K^2)}{n_y h^d}} + \sqrt{\frac{2\log(1/\delta)}{n_y}} \right.$$
$$\left. + \left| \begin{array}{l} \phi_y h^2 \quad \text{under Assumptions 4 and 5} \\ \psi_y h \quad \text{if } h \le r_0 \quad \text{under Assumptions 6 and 7} \end{array} \right. \right\}$$

has probability at least $1 - \delta$ conditionally on $Y_{1:n}$ and when $n_y > 0$, which is the result. $\qquad\square$

**Proposition 8** ($L^1-$bound on kernel density estimators based on KDEO). *Let $\delta \in (0, 1/2)$. Then, on the event $\{n_1 > 0\}$, we have*

$$\mathbb{P}\left( \int |\hat{f}_{1s}^*(x) - f_1(x)|dx \le \hat{c}_{1,d,\varepsilon}(1+s^{(d+\varepsilon)/2})\sqrt{\frac{M_0(K^2)}{ms^d}} \right.$$
$$+ c_{1,d,\varepsilon}(1+h^{(d+\varepsilon)/2})\sqrt{\frac{M_0(K^2)}{n_1 h^d}}$$
$$+ \sqrt{2\log(1/\delta)}\left(\frac{1}{\sqrt{m}} + \frac{1}{\sqrt{n_1}}\right)$$
$$\left. + \left\{ \begin{array}{l} \phi_1(h^2+s^2) \quad \text{under Assumptions 4 and 5} \\ \psi_1(h+s) \quad \text{if } h,s \le r_0 \text{ under Assumptions 6 and 7} \end{array} \right. \middle| Y_{1:n} \right)$$
$$\ge 1 - 2\delta$$

*where $\hat{c}_{1,d,\varepsilon} = C_{d,\varepsilon}(1 + \sqrt{M_{d+\varepsilon}(\hat{f}_{1h}+K)})$ and with the notation of Proposition 7.*

*Proof.* Write the quantity of interest as

$$\int |\hat{f}_{1s}^*(x) - f_1(x)|dx \le \int |\hat{f}_{1s}^*(x) - \mathbb{E}(\hat{f}_{1s}^*(x)|\mathcal{D}_n)|dx + \int |\mathbb{E}(\hat{f}_{1s}^*(x)|\mathcal{D}_n) - f_1(x)|dx. \tag{19}$$

We control the two terms on the right-hand side separately. Define

$$T_{\mathcal{D}_n}(X_1^*,\ldots,X_m^*) = \int |\hat{f}_{1s}^*(x) - \mathbb{E}(\hat{f}_{1s}^*(x)|\mathcal{D}_n)|dx\,1_{\{n_1>0\}}$$

and note that the function $T_{\mathcal{D}_n}$ satisfies the assumption of Lemma 2 with $C_i = (2/m)1_{\{n_1>0\}}$. Since, given $\mathcal{D}_n$ and $\{n_1 > 0\}$, the $X_{1i}^*$ are i.i.d. generated according to $\hat{f}_{1h}$, we can therefore apply the McDiarmid inequality, conditionally on $\mathcal{D}_n$, to obtain, with probability at least $1 - \delta$ (conditionally on $\mathcal{D}_n$),

$$\int |\hat{f}_{1s}^*(x) - \mathbb{E}(\hat{f}_{1s}^*(x)|\mathcal{D}_n)|dx \le \int \mathbb{E}\left[\left|\hat{f}_{1s}^*(x) - \mathbb{E}(\hat{f}_{1s}^*(x)|\mathcal{D}_n)\right| \middle| \mathcal{D}_n\right]dx + \sqrt{\frac{2\log(1/\delta)}{m}}1_{\{n_1>0\}}.$$

Then the Jensen inequality gives that, with probability at least $1 - \delta$ (conditionally on $\mathcal{D}_n$),

$$\int |\hat{f}_{1s}^*(x) - \mathbb{E}(\hat{f}_{1s}^*(x)|\mathcal{D}_n)|dx \leq \int \sqrt{\mathbb{E}\left[\left(\hat{f}_{1s}^*(x) - \mathbb{E}(\hat{f}_{1s}^*(x)|\mathcal{D}_n)\right)^2 |\mathcal{D}_n\right]}dx$$
$$+ \sqrt{\frac{2\log(1/\delta)}{m}}1_{\{n_1>0\}}.$$

Now we investigate the behavior of the first term in the right-hand side on the event $\{n_1 > 0\}$. We have

$$\mathbb{E}\left[\left(\hat{f}_{1s}^*(x) - \mathbb{E}(\hat{f}_{1s}^*(x)|\mathcal{D}_n)\right)^2 |\mathcal{D}_n\right] = \frac{1}{m}\text{Var}(K_s(x - X_{11}^*)|\mathcal{F}_n) \leq \frac{1}{m}\mathbb{E}(K_s^2(x - X_{11}^*)|\mathcal{F}_n)$$
$$= \frac{M_0(K^2)}{ms^d}\hat{f}_{1h} * \tilde{K}_s(x)$$

with $\tilde{K} = K^2/M_0(K^2)$. Using Lemma 5 and the previous probability bound, we get that the event

$$E_1 = \left\{\int |\hat{f}_{1s}^*(x) - \mathbb{E}(\hat{f}_{1s}^*(x)|\mathcal{D}_n)|dx \leq \hat{c}_{1,d,\varepsilon}(1 + s^{(d+\varepsilon)/2})\sqrt{\frac{M_0(K^2)}{ms^d}} + \sqrt{\frac{2\log(1/\delta)}{m}}\right\}$$

satisfies $\mathbb{P}(E_1^c|\mathcal{D}_n)1_{\{n_1>0\}} \leq \delta$, and then, integrating out the conditional expectation with respect to $X_1, \ldots, X_n$,

$$\mathbb{P}(E_1^c|Y_{1:n})1_{\{n_1>0\}} \leq \delta. \tag{20}$$

To control the second term in (19), write

$$\int |\mathbb{E}(\hat{f}_{1s}^*(x)|\mathcal{D}_n) - f_1(x)|dx = \int |\hat{f}_{1h} * K_s(x) - f_1(x)|dx$$
$$\leq \int |(\hat{f}_{1h} - f_1) * K_s(x)|dx + \int |f_1 * K_s(x) - f_1(x)|dx$$
$$\leq \int |\hat{f}_{1h}(x) - f_1(x)|dx + \int |f_1 * K_s(x) - f_1(x)|dx.$$

It was shown at the end of Proposition 7 that the event

$$\left\{\int |\hat{f}_{1h}(x) - f_1(x)|dx \leq c_{1,d,\varepsilon}(1 + h^{(d+\varepsilon)/2})\sqrt{\frac{M_0(K^2)}{n_1 h^d}} + \sqrt{\frac{2\log(1/\delta)}{n_1}}\right.$$
$$\left. + \left|\begin{array}{l}\phi_1 h^2 \text{ under Assumptions 4 and 5} \\ \psi_1 h \text{ if } h \leq r_0 \text{ under Assumptions 6 and 7}\end{array}\right.\right\}$$

has probability at least $1 - \delta$ conditionally on $Y_{1:n}$ and when $n_1 > 0$. This statement with Lemma 6 under Assumptions 4-5, or Lemma 7 under Assumptions 6-7 when $s \leq r_0$, allows to obtain that the event

$$E_2 = \left\{\int |\mathbb{E}(\hat{f}_{1s}^*(x)|\mathcal{D}_n) - f_1(x)|dx \leq c_{1,d,\varepsilon}(1 + h^{(d+\varepsilon)/2})\sqrt{\frac{M_0(K^2)}{n_1 h^d}} + \sqrt{\frac{2\log(1/\delta)}{n_1}}\right.$$
$$\left. + \left|\begin{array}{l}\phi_1(h^2 + s^2) \text{ under Assumptions 4 and 5} \\ \psi_1(h + s) \text{ if } h, s \leq r_0 \text{ under Assumptions 6 and 7}\end{array}\right.\right\}$$

satisfies

$$\mathbb{P}(E_2^c|Y_{1:n})1_{\{n_1>0\}} \leq \delta. \tag{21}$$

A combination of (19), (20) and (21) yields

$$\mathbb{P}(E_1^c \cup E_2^c|Y_{1:n})1_{\{n_1>0\}} \leq \mathbb{P}(E_1^c|Y_{1:n})1_{\{n_1>0\}} + \mathbb{P}(E_2^c|Y_{1:n})1_{\{n_1>0\}} \leq 2\delta.$$

The result follows immediately. $\qquad\square$

*End of the proof of Theorem 6.* Since $m = n_0 1_{\{n_1 > 0\}}$, we clearly have $\hat{\eta}^*(x) > 1/2$ if and only if $\hat{f}_{1s}^*(x) > \hat{f}_{0s}(x)$. Hence we can apply Theorem 5 to obtain

$$R_{1/2}(\hat{g}^*) - R_{1/2}(g) \leq \frac{1}{2} \int |\hat{f}_{0s}(x) - f_0(x)| dx + \frac{1}{2} \int |\hat{f}_{1s}^*(x) - f_1(x)| dx.$$

Using finally Proposition 7 to control the first term and Proposition 8 to control the second one, we conclude that

$$\mathbb{P}\Bigg( R_{1/2}(\hat{g}^*) - R_{1/2}(g)$$

$$\leq \frac{c_{0,d,\varepsilon}}{2}(1 + s^{(d+\varepsilon)/2})\sqrt{\frac{M_0(K^2)}{n_0 s^d}} + \frac{c_{1,d,\varepsilon}}{2}(1 + h^{(d+\varepsilon)/2})\sqrt{\frac{M_0(K^2)}{n_1 h^d}}$$

$$+ \frac{\hat{c}_{1,d,\varepsilon}}{2}(1 + s^{(d+\varepsilon)/2})\sqrt{\frac{M_0(K^2)}{n_0 s^d}} + \sqrt{2\log(1/\delta)}\left(\frac{1}{\sqrt{n_0}} + \frac{1}{2\sqrt{n_1}}\right)$$

$$+ \frac{1}{2}\begin{cases} \phi_0 s^2 + \phi_1(h^2 + s^2) & \text{under Assumptions 4 and 5} \\ \psi_0 s + \psi_1(h + s) & \text{if } h, s \leq r_0 \text{ under Assumptions 6 and 7} \end{cases} \Bigg| Y_{1:n} \Bigg) 1_{\{n_0 n_1 > 0\}} \geq 1 - 3\delta.$$

Rearrange the above upper bound (taken to be infinite on the event $\{n_0 n_1 = 0\}$, and hence also valid on this event) and integrate out the conditional expectation given $Y_{1:n}$ to complete the proof. $\qquad\square$

## G  Analysis of the kernel smoothing plug-in rule

Let us finally highlight that we get, as a direct byproduct of Theorem 5 and Proposition 7, the following bound on the risk of the kernel discrimination rule

$$\hat{g}(x) = \hat{g}_h(x) = 1_{\{\hat{f}_{1h}(x) > \hat{f}_{0h}(x)\}}$$

based on the initial data only. This will be used for comparison purposes with Theorem 6 in Remark 5.

**Proposition 9.** *Let $\delta \in (0, 1/2)$. Then, with probability at least $1 - 2\delta$,*

$$R_{1/2}(\hat{g}) - R_{1/2}(g) \leq \frac{\sqrt{M_0(K^2)}}{2}\left(\frac{c_{0,d,\varepsilon}}{\sqrt{n_0 h^d}} + \frac{c_{1,d,\varepsilon}}{\sqrt{n_1 h^d}}\right)(1 + h^{(d+\varepsilon)/2})$$

$$+ \sqrt{\frac{\log(1/\delta)}{2}}\left(\frac{1}{\sqrt{n_0}} + \frac{1}{\sqrt{n_1}}\right)$$

$$+ \frac{1}{2}\begin{cases} (\phi_0 + \phi_1)h^2 & \text{under Assumptions 4 and 5} \\ (\psi_0 + \psi_1)h & \text{if } h \leq r_0 \text{ under Assumptions 6 and 7} \end{cases}$$

*with the notation of Proposition 7, where the upper bound is taken to be infinite when $n_0 n_1 = 0$.*

## H  Additional numerical results

### H.1  Algorithms for oversampling methods

Algorithm 1 explains the different steps of the SMOTE algorithm for generating synthetic samples while highlighting the different hyperparameters involved.

KDE-based oversampling (KDEO) is another option to tackle the imbalanced data in a similar spirit to SMOTE. In contrast with SMOTE, the KDEO algorithm relies on a kernel function $K$ and a bandwidth matrix $H$ that governs the smoothness of the estimated density function and the tradeoff between bias and variance. Since the parameter $H$ has a significant impact on the accuracy of KDE, several methods have been developed to obtain an appropriate choice of $H$. Algorithm 2 explains the steps KDEO follows for oversampling in compliance with the bandwidth selected through the multivariate version of Scott's rule of thumb [Scott, 2015].

---

**Algorithm 1** SMOTE

---

**Input:** Samples $\{X_{11}, \ldots, X_{1n_1}\} \subset \mathbb{R}^d$, number of nearest neighbors $k \in \{0, 1, \ldots, n_1 - 1\}$, and number of synthetic samples $m$

1: **for** each $i = 1, \ldots, m$ **do**

    Generate $\tilde{X}_{1i}$ uniformly among $\{X_{1i}\}_{1 \le i \le n_1}$.

    If $n_1 > 1$ and $k > 0$, generate $\overline{X}_{1i}$ uniformly among the $k$ nearest neighbors to $\tilde{X}_{1i}$ in the minority class deprived of $\tilde{X}_{1i}$. Else, do $\overline{X}_{1i} = \tilde{X}_{1i}$.

    $X_{1i}^* = (1 - \lambda)\tilde{X}_{1i} + \lambda\overline{X}_{1i}$ , where $\lambda \sim \mathcal{U}[0, 1]$.

2: **end for**

3: Return $(X_{1i}^*, \ldots, X_{1m}^*)$

---

**Algorithm 2** KDEO

---

**Input:** Samples $\{X_{11}, \ldots, X_{1n_1}\} \subset \mathbb{R}^d$, number of synthetic samples $m$

1: Compute the (empirical) covariance matrix $S$ of minority samples and set the bandwidth matrix $H_1$ to satisfy $H_1^2 = n_1^{-2/(d+4)} S$ (Scott's rule of thumb).

2: **for** each $i = 1, \ldots, m$ **do**

    Generate $\tilde{X}_{1i}$ uniformly among $\{X_{1i}\}_{1 \le i \le n_1}$.

    Generate $W_i \sim \mathcal{N}(0, I_p)$.

    Generate $X_{1i}^* = \tilde{X}_{1i} + H_1 W_i$.

3: **end for**

4: Return $(X_{1i}^*, \ldots, X_{1m}^*)$

---

## H.2    Simulated data

We provide results linked to the analysis of Examples 1 and 4 in the main paper. Before that, we also introduce a further example, where again, the $\{(X_i, Y_i)\}_{1 \le i \le n}$ are $n = 1000$ i.i.d. samples from the distribution of $(X, Y)$ and $e_i$ is the $i$th vector in the canonical basis of $\mathbb{R}^d$.

**Example S.1:** Let $Z = \mathcal{B}Y_1 + (1 - \mathcal{B})Y_2$, with $\mathcal{B} \sim \text{Bernoulli}(0.5)$, $Y_1$ is an extended generalized Pareto random variable, i.e. $Y_1 \sim \text{EGPD}(\kappa(X), \sigma(X), 0.5)$ and $Y_2 \sim \text{Exp}(10\gamma(X))$, where $\kappa(X) = \exp(X^\top e_1)$, $\sigma(X) = \exp(X^\top e_2)$ and $\gamma(X) = \exp(X^\top e_3)$. Define $Y = 1_{\{Z > t\}}$ with $t$ tuning class imbalance.

In Example S.1, the parameter $t$ is tuned to achieve various probability levels $(1 - p_1 = 0.60, 0.80, 0.85, 0.90, 0.92, 0.95)$. The considered methods are the ones introduced in Sections 3.2 and 4.1 of the main document.

The AM-risk performances of the KS and $K$NN classifiers for Examples 1, 4, and S.1 are reported in Figure 6, where both methods exhibit identical patterns and achieve superior performance with KDEO and BBC compared to SMOTE. Moreover, the $K$NN classifier can further be improved by tuning the hyperparameter $K$ via CV, as shown in Figure 7; A noticeable performance improvement can be observed in the KDEO-CV, SMOTE-CV results, but CV on $K$ yields poorer performance for the BBC. Moreover, SMOTE-CV, KDEO and BBC exhibit very similar performance, indicating that these methods enhance minority class representation comparably when used with $K$NN, which is consistent with the theoretical justifications presented in the main document.

We also consider other classifiers, i.e. Random Forest and Logistic Regression (LR), compared to the the main document. Figure 8 shows the AM-risk results for the Random Forest classifier. The KDEO and BBC perform better than SMOTE in terms of AM-risk. Further notice that KDEO still performs better than BBC even when the probability of the minority class $p_1$ is small. In Example 4, all methods perform similarly. We did not use CV to select the number of features at each split, as the low dimensionality makes CV not beneficial; without CV, BBC performs comparably to the oversampling methods across all examples.

In contrast, the LR classifier performs uniformly across all resampling methods (see Figure 9). Notice that the LR classifier performs worse when applied to more complex classification structures. This overall uniformity in results is in line with the results given in Theorems 1 and 3 as they suggest that

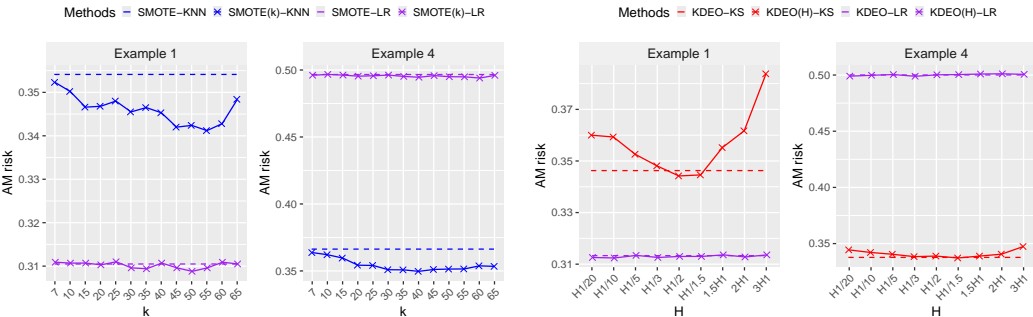

Figure 5: Average AM-risk of KNN, KS, and LR classifiers on balanced data over 50 replications. *Left:* using SMOTE and SMOTE($k$) with $k \in (7, 65)$. *Right:* using KDEO and KDEO($H$), with $H = cH_1$ and $c$ ranging in $(1/20, 3)$ where $H_1$ follows from Scott's rule.

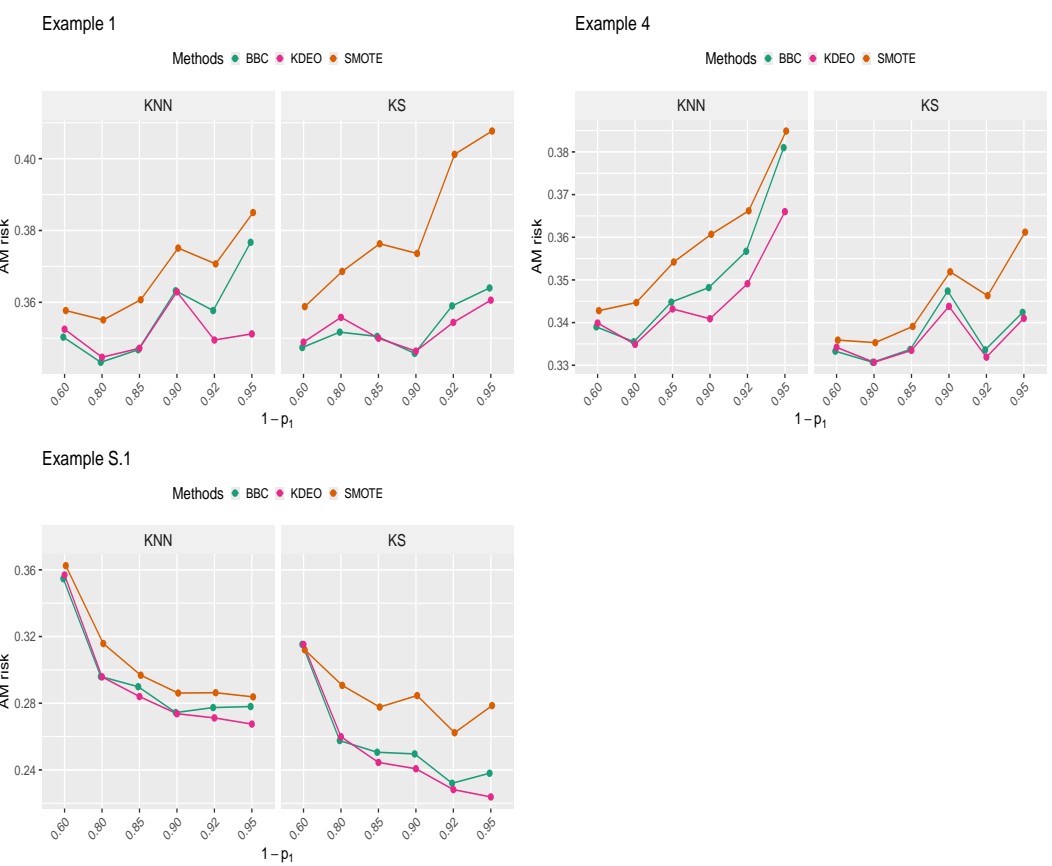

Figure 6: Average AM-risk across different data imbalance regimes for the KS (described in Section 3.2) and $K$NN classification rules computed over 50 replications.

estimating the risk with SMOTE or KDEO gives better results when $K$ and $H$ are small, underlining that oversampling may not be critical for such type of (parametric) classifier.

## H.3    Real data analysis: Abalone, California, MagicTel, Phoneme, and House_16H datasets

We apply the Random Forest, KS, LR, and LR-Lasso classifiers to the real datasets considered in the main paper. The AM-risk results for the KS are given in Figure 10, those for Random Forest in Figure 11, while results for the LR and LR-Lasso classifiers are presented in Figure 12. The KS

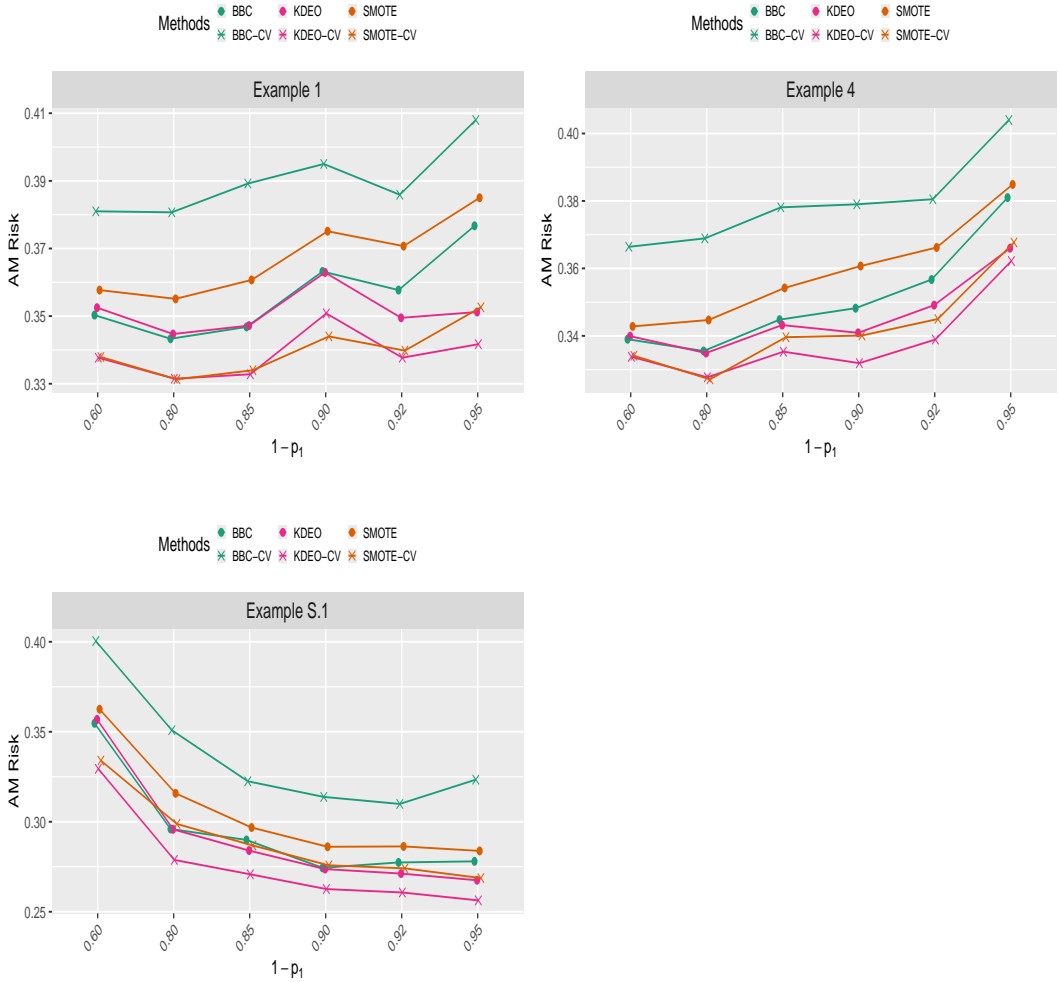

Figure 7: Average AM-risk across different data imbalance regimes for the $K$NN methods described in Section 4.1, computed over 50 replications.

classifier shows comparable performance to SMOTE and KDEO across all datasets. For instance, in the MagicTel dataset, KDEO performs slightly better, while California and Abalone SMOTE stand superior. For the Random Forest classifier, SMOTE and KDEO-based oversampling exhibit comparable performance across all datasets. Similarly, both oversampling methods yield nearly identical results in the LR and LR-Lasso classifiers, except for the Abalone and House_16H datasets. A key finding is that using cross-validation to select the number of features at each split in Random Forest and the regularization parameter $\lambda$ in LR-Lasso consistently enhances performance across all scenarios, with the exception of Abalone and House_16H. The low dimensionality of the Abalone and House_16H datasets may explain the lack of performance improvement when CV is used for parameter tuning.

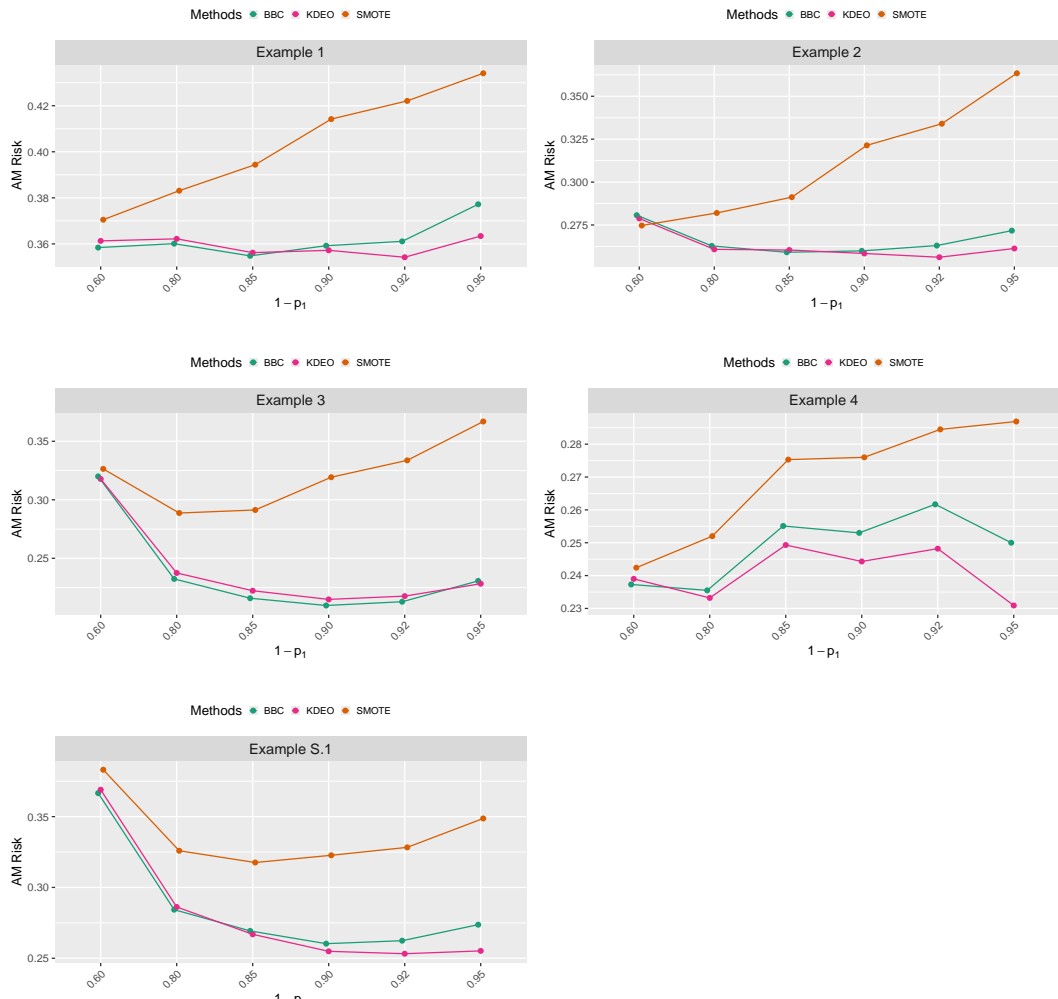

Figure 8: Average AM-risk across different data imbalance regimes for the Random Forest classifier computed over 50 replications.

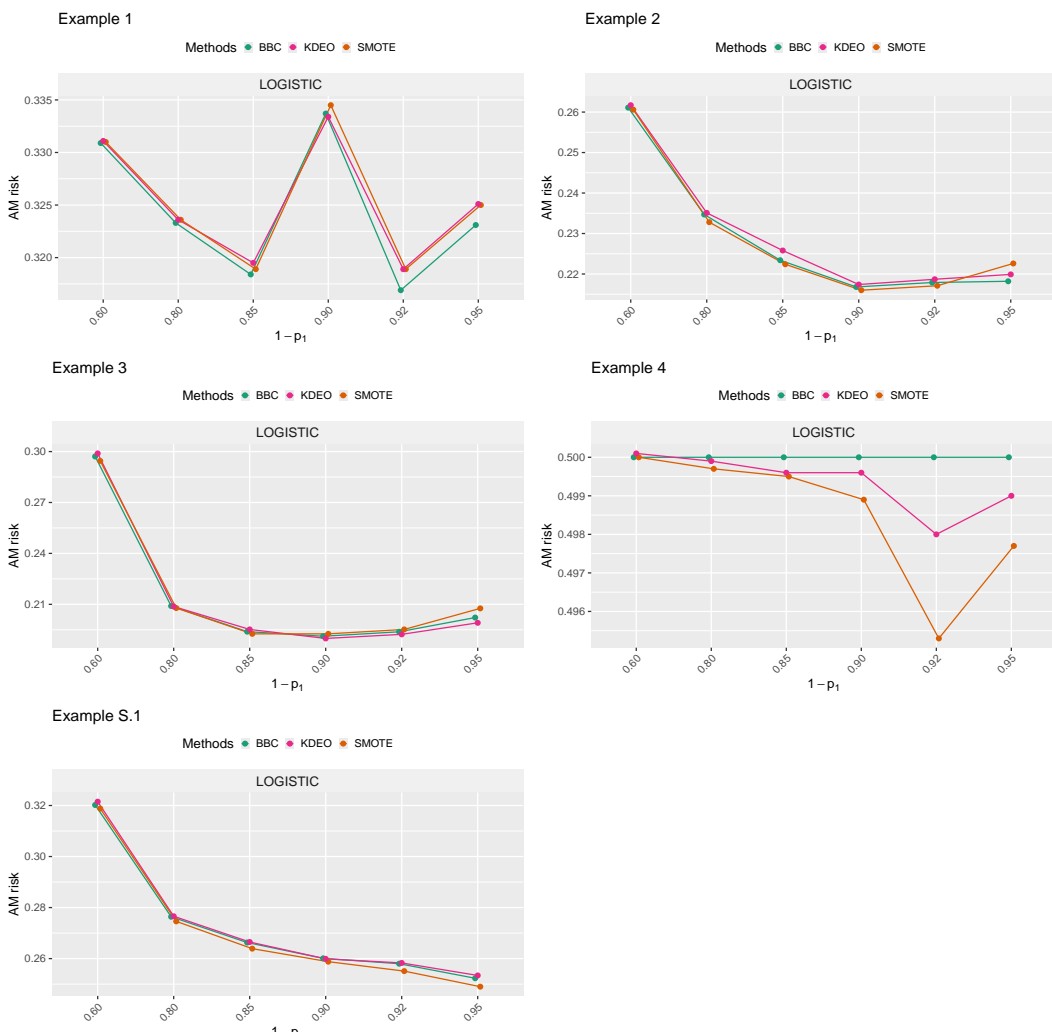

Figure 9: Average AM-risk across different data imbalance regimes for the Logistic Regression classifier computed over 50 replications.

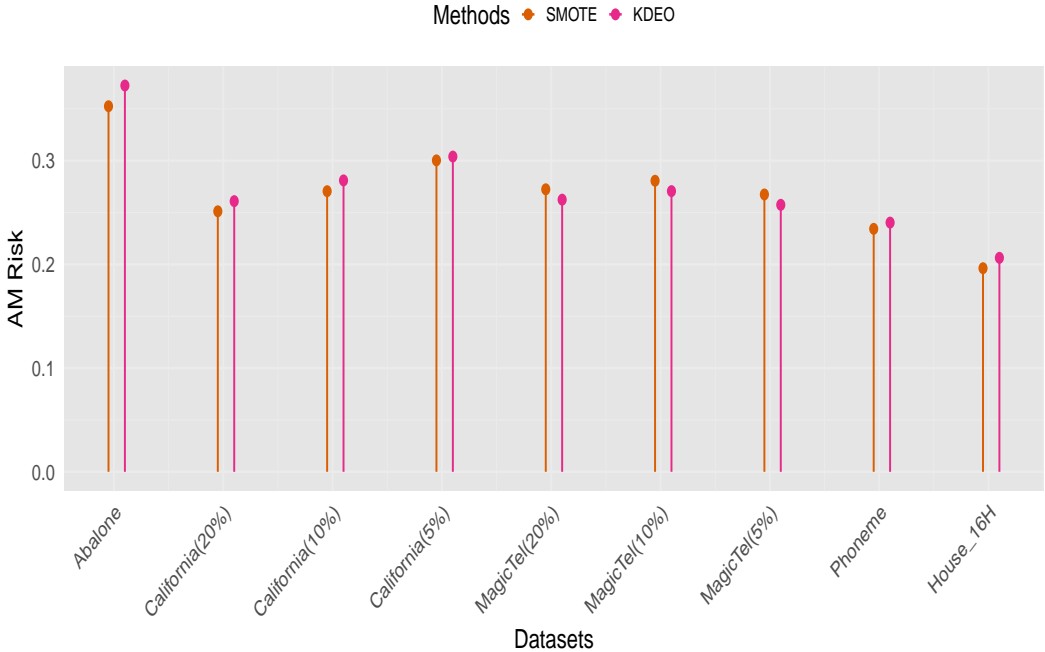

Figure 10: AM-risk corresponding to different rebalancing methods and datasets when using the KS classifier.

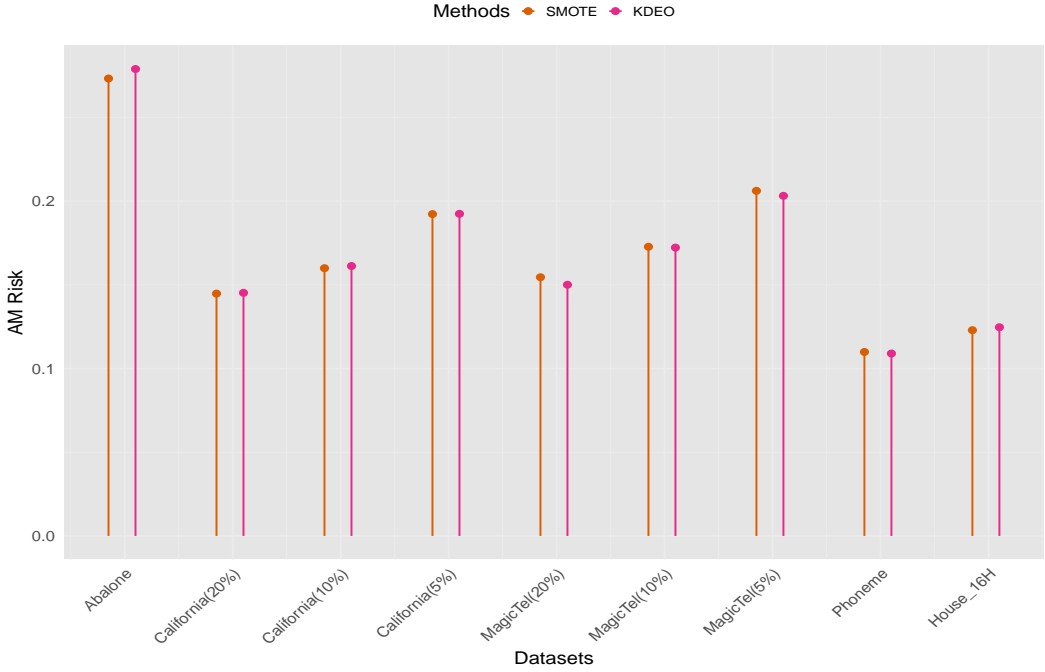

Figure 11: AM-risk corresponding to different rebalancing methods and datasets when using the Random Forest classifier.

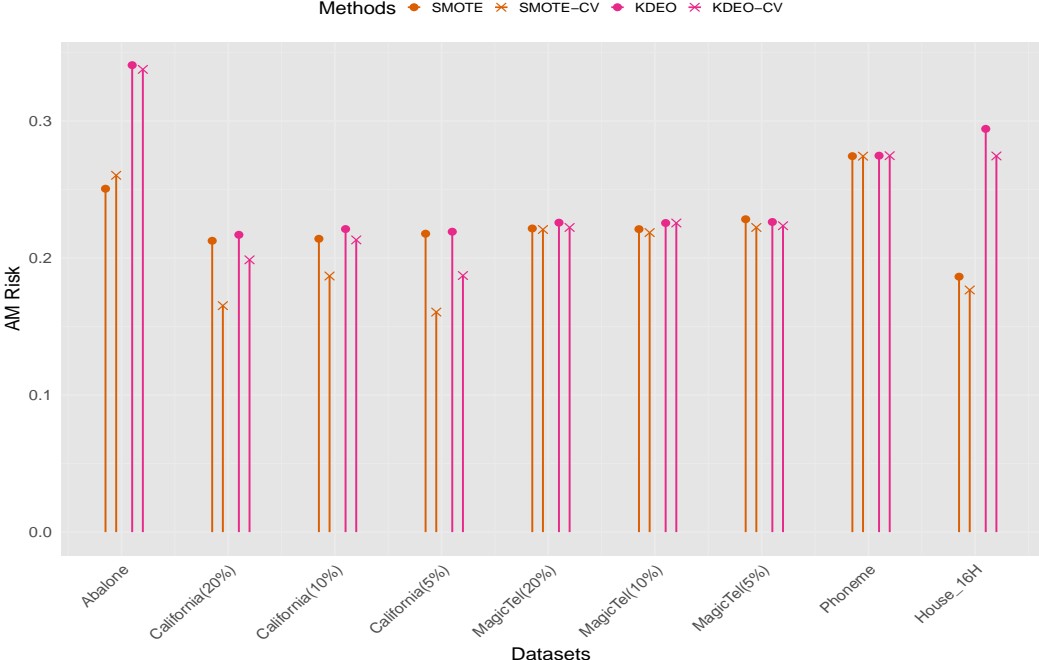

Figure 12: AM-risk corresponding to different rebalancing methods and datasets when using the LR and LR-Lasso classifier. Methods containing the "CV" characters stand for the LR-Lasso classifier with a regularization parameter tuned through CV.

