# OpenReview forum: "Concentration and excess risk bounds for imbalanced classification with synthetic oversampling"
_NeurIPS.cc/2025/Conference — NeurIPS 2025 poster_

### Official Review · Reviewer_eFiF · 2025-06-23

**Clarity:** 3
**Significance:** 3
**Originality:** 3
**Rating:** 4
**Confidence:** 2

**Summary:**

This paper presents a rigorous theoretical framework for analyzing SMOTE and KDE-based oversampling techniques in imbalanced classification. The work makes significant theoretical contributions while maintaining practical relevance.

**Questions:**

How does performance scale with increasing dimensionality?

**Ethical Concerns:**

["NO or VERY MINOR ethics concerns only"]

**Limitations:**

The author has established important theoretical results in the paper and should discuss how to guide the selection of hyperparameters in practical applications.

**Quality:**

3

**Strengths And Weaknesses:**

**Strengths:**

1. Theoretical contributions. Fills important theoretical gaps in understanding SMOTE and its variants and provides mathematical foundation for widely-used oversampling techniques.
2. Extends classical non-parametric risk analysis to imbalanced settings.
3. Comprehensive numerical experiments support theoretical findings.

**Weaknesses:**
1. Enhanced readability. For readers with limited mathematical background, the paper can improve: more intuitive explanations of key concepts, intuitive interpretations of theorems, and visualizations of the theoretical relationships.
2. Expanded experimental analysis. Evaluation on a broader range of datasets, especially extremely imbalanced cases and analysis of scalability with increasing dimensionality.
4. Lack conclusion. The paper ends directly after the experimental part without conclusion.

---

> ### Author Rebuttal · Authors · 2025-07-29
>
> First of all, we would like to thank the referee for his or her valuable comments and suggestions, which will help us to improve the quality and clarity of the manuscript. In what follows we provide answers to all of his or her comments suggesting the  changes that we would like to do in the next version of our paper.
>
> **1) Enhanced readability**
>
> Thank you for the reviewer’s suggestion. In the revised version, we will include more intuitive explanations of the key concepts and main results, along with some illustrative examples.
>
> **2) Expanded experimental analysis**
>
> We are happy to explore additional datasets with higher dimensions and to expand our experimental results.
>
> **3) Lack conclusion**
>
> We will add a conclusion section and summarize the key contributions in the revised version.
>
> **Questions**
>
> **How does performance scale with increasing dimensionality?**
>
> In our theoretical bounds, the variance term—which decreases at the rate of $\frac{1}{\sqrt{n_{1}}}$—is independent of the dimension, whereas the bias term slows with increasing dimension due to the presence of the factor $\left( \frac{k}{n_{1}} \right)^{1/d}$. As $d $ increases, the bias decays more slowly and eventually dominates the variance. In practice, when the dimensionality is too high, one can apply PCA as a preprocessing step or use manifold-based SMOTE to mitigate the curse of dimensionality.
>
> **Limitations**
>
> **The author has established important theoretical results in the paper and should discuss how to guide the selection of hyperparameters in practical applications.**
>
> We selected the optimal values for these hyperparameters, as shown in Figure 1, where the effects of varying $h$ and $k$ under Theorem 4 are delineated. Cross-validation was also employed to tune the classifier's hyperparameters, resulting in a significant performance improvement, as illustrated in Figure 2. We will investigate this further in the revised manuscript.

---

> > ### Comment · Reviewer_eFiF · 2025-08-03
> >
> > Thank the authors for the rebuttal. After reading the rebuttal and other reviews, I tend to maintain my original score.

---

### Official Review · Reviewer_P1wd · 2025-07-03

**Clarity:** 3
**Significance:** 2
**Originality:** 3
**Rating:** 4
**Confidence:** 3

**Summary:**

This paper provides a theoretical framework for analyzing SMOTE and its variants, widely used for imbalanced classification. It establishes an exponential inequality bounding the gap between empirical and population risk on synthetic data, and proves that kernel-based classifiers trained on such data can achieve minimax optimal rates. The results offer practical insights for tuning SMOTE and downstream models, supported by numerical experiments.

**Questions:**

1. How do the theoretical guarantees extend to or break down when using modern non-kernel classifiers, such as deep neural networks, which are commonly used in imbalanced classification tasks?

2. Can the authors comment on the robustness of their bounds in high-dimensional settings, where synthetic samples generated by SMOTE may suffer from over-smoothing or introduce unrealistic interpolations?

3. How does the proposed framework handle or adapt to more advanced oversampling strategies (e.g., BorderlineSMOTE, ADASYN), and can similar concentration and excess risk bounds be established for those methods?

**Ethical Concerns:**

["NO or VERY MINOR ethics concerns only"]

**Final Justification:**

I find the potential extension of the SMOTE concentration result to a uniform concentration inequality over a class of functions particularly interesting, and I encourage the authors to include a discussion of this. My concerns have been largely addressed, and I will maintain my current evaluation.

**Limitations:**

Yes.

**Paper Formatting Concerns:**

N/A.

**Quality:**

3

**Strengths And Weaknesses:**

Strengths:

1. The paper establishes an exponential concentration inequality quantifying the discrepancy between empirical and population risk on synthetic minority samples. It also proves minimax optimality of kernel-based classifiers trained with SMOTE-generated data, an important contribution that formalizes a widely used heuristic.

2. The theoretical insights directly inform practical guidelines for tuning SMOTE parameters and the downstream classifier, enhancing the usability of the findings.

3. Establishing minimax rates when learning from synthetic examples is a fresh angle in theoretical ML and adds depth to the understanding of oversampling strategies.

Weaknesses:

1. The theoretical results rely on smoothness assumptions and kernel-based classifiers, which may not capture the behavior of neural networks commonly used in practice.

2. The framework appears focused on binary classification. An extension to multi-class imbalanced settings would broaden applicability and address a more general class of real-world problems.

---

> ### Author Rebuttal · Authors · 2025-07-29
>
> First of all, we would like to thank the referee for his or her valuable comments and suggestions, which will help us to improve the quality and clarity of the manuscript. In what follows we provide answers to all of his or her comments suggesting the  changes that we would like to do in the next version of our paper.
>
> **1) The theoretical results rely on smoothness assumptions and kernel-based classifiers ... :**
>
> Theorem 1 does not rely on smoothness assumptions and only depends on Lipschitz continuity, which includes the case of neural networks. However, it does not provide an excess risk bound for a classifier trained on synthetically generated data.
>
> Furthermore, we investigated possible extensions of the SMOTE concentration result. Encouragingly, we were able to generalize it to a uniform concentration inequality over a class of functions, yielding an excess risk bound for SMOTE. The resulting bound is similar in structure to the original one, with the addition of a Rademacher complexity term to account for the function class's richness. This extension highlights the relevance of our results to neural network learning with SMOTE, and we consider it an important theoretical contribution.
>
> This new uniform convergence bound has the same form as Theorem 1, except that it holds uniformly over a class $\mathcal{F}$ of $L$-Lipschitz functions that are uniformly bounded, i.e., $\sup_{G \in \mathcal{F}} \|G\|_{\infty} \leq B$. The bound on the supremum of the difference between the empirical mean and the population mean
>
> $\sup_{G\in \mathcal{F}}\left|\frac{1}{m} \sum_{i=1}^{m} G (X_{1i}^{*}) - \mathbb{E}_{1}[G(X)]\right|$
>
>  remains structurally the same, with the addition of a Rademacher complexity term corresponding to the function class $\mathcal{F}$. Note that it does not involves too much changes in view of the revision.
>
> The proof is similar to that of the previous version, except that we now use the Talagrand--Bousquet inequality, which provides a concentration bound for the quantity $S_n - \mathbb{E}[S_n]$, where
>
> $$S_n = \sup_{f \in \mathcal{F}} \sum_{i=1}^n f(Z_i),$$
>
> the random variables $Z_i$ are i.i.d. random variables, and $\mathcal{F}$ is a uniformly bounded class of functions.
>
> **2) The framework appears focused on binary classification.**
>
> Theorems 1 and 2 do not, in fact, assume a binary classification setting; rather, they show that the empirical mean with respect to the probability of a synthetically-generated class concentrates around the population mean. This is of course readily adapted to the multi-class classification setting by fixing one reference class from which synthetic observations are sampled and as such, our results immediately apply to multi-class classification. However, the current statements of Theorems 1 and 2 do not provide excess risk bounds for a learned classifier. With the extension of these results to the uniform convergence setting, we can obtain excess risk bounds for multi-class classification as well, as explained in the previous answer.
>
> **Questions :**
>
> **1) How do the theoretical guarantees extend to or break down when using modern non-kernel classifiers**
>
> As mentioned in the previous question, since we have established a uniform convergence result for a class of uniformly bounded functions, it implies that we can derive excess risk bounds for classifiers such as neural networks, given their finite Rademacher complexities. This extension directly addresses the reviewer’s question and will be incorporated into the revised version.
>
> **2) Can the authors comment on the robustness of their bounds in high-dimensional settings**
>
> Our bounds remain valid in any dimension. As the dimension $d$ increases, the bias term $(k/n_{1})^{1/d}$ shrinks more slowly and can overtake the $n_{1}^{-1/2}$ variance term, so the overall upper bound becomes looser—the classic curse of dimensionality. Using a moderate neighborhood size (e.g., $k \propto n_{1}^{4/(d+4)}$ ) or first projecting onto a lower–dimensional subspace (PCA or manifold‑SMOTE) keeps this bias under control and prevents oversmoothing from dominating.
>
> **3) How does the proposed framework handle or adapt to more advanced oversampling strategies**
>
> The main difficulty with these approaches is that they select neighborhoods from both classes. In contrast, we rely on uniform bounds for k-NN, which are based on data from a single distribution. In the case of BorderlineSMOTE, one must handle data from two distributions simultaneously while still bounding the risk with respect to one of them.

---

> > ### Comment · Reviewer_P1wd · 2025-08-05
> >
> > Thank you. I find the potential extension of the SMOTE concentration result to a uniform concentration inequality over a class of functions particularly interesting, and I encourage the authors to include a discussion of this. My concerns have been largely addressed, and I will maintain my current evaluation.

---

### Official Review · Reviewer_pe2f · 2025-07-03

**Clarity:** 3
**Significance:** 3
**Originality:** 3
**Rating:** 5
**Confidence:** 4

**Summary:**

This paper provides a theoretical analysis of synthetic oversampling methods—specifically SMOTE and KDE-based techniques—for binary classification under severe class imbalance. The authors analyze the statistical properties of oversampling from both a concentration and excess risk perspective:
	1.	Concentration bounds (Theorems 1 & 2, pages 5–6) are derived to relate the empirical risk computed on synthetic minority samples to the true population risk. These are obtained for both SMOTE (via nearest neighbors) and KDE (via random samples from an estimated density).
	2.	Excess AM-risk bound (Theorem 4, page 11) is established for plug-in classifiers trained on synthetic data generated via KDE. The bound decomposes the error into bias and variance components and suggests optimal bandwidths for both sample generation and classifier construction.
	3.	Practical guidance is derived from the theory—e.g., justifying small k in SMOTE and the importance of tuning bandwidths in KDE—especially under extreme imbalance n_1/n_0 \to 0.
	4.	Experiments on both synthetic and real datasets (Section 5) validate these theoretical findings and examine the effect of different oversampling hyperparameters on performance across various classifiers.

The work unifies existing data-level and model-level approaches to class imbalance, offers new non-asymptotic theoretical tools, and motivates principled hyperparameter selection in oversampling pipelines.

**Questions:**

1.	Extension of Theorem 4 to SMOTE:
You mention (line 257) that excess risk analysis for SMOTE is more difficult. Is this due to the discontinuous support of the interpolated samples? Could an approximation (e.g., under smoothness of f^*) yield a rough bound?
	2.	Bandwidth mismatch:
Theorem 4 (line 237) suggests that the optimal KDE bandwidth for sample generation may differ from the optimal classifier bandwidth. Do your experiments support this? Can you comment on how sensitive the AM-risk is to this mismatch?
	3.	Logistic regression stability:
In line 335 (page 14), you mention that logistic regression is relatively unaffected by oversampling. Given that Theorems 1–2 still apply in principle, is this due to insensitivity to data augmentation, or model robustness to density perturbation?
	4.	Applicability of concentration bounds to learned classifiers:
Would it be possible to extend Theorems 1–2 to uniform convergence over a class of Lipschitz classifiers (e.g., KNN or kernel classifiers)? If not, could empirical Rademacher complexity or covering number arguments be layered on top?

Suggestions for Improvement
	•	Add a brief discussion on how Assumption 5 aligns with the geometry of P(X|Y=1). If the data lives in \mathbb{R}^d, the measure condition should reflect dimensionality.
	•	Consider including a KDE vs. SMOTE comparison in excess risk terms, even if only heuristic or simulated.
	•	Include a toy experiment showing the effect of varying h and s in accordance with Theorem 4, to highlight the practical relevance of the theory.
	•	Clarify whether your theoretical guidance can improve hyperparameter tuning in practice compared to cross-validation.

**Ethical Concerns:**

["NO or VERY MINOR ethics concerns only"]

**Final Justification:**

I've read the responses and the comments, I've decided to keep my score.

**Limitations:**

Yes

**Quality:**

3

**Strengths And Weaknesses:**

Strengths
	•	Novel theoretical analysis: The concentration bounds (Theorems 1 & 2) and excess risk result (Theorem 4) represent significant theoretical contributions that go beyond standard margin-based generalization theory. The analysis draws from nonparametric statistics and covers both SMOTE-like (nearest-neighbor-based) and kernel-based oversampling.
	•	Practical motivation and implications: The work sheds light on popular heuristics such as SMOTE with k=5, showing when such defaults are suboptimal. The theoretical results suggest parameter tuning strategies, particularly in severely imbalanced regimes.
	•	Unifying perspective: The framing of synthetic oversampling as a reweighting scheme that alters the data distribution while maintaining the form of the classifier allows the authors to bridge the gap between model-level and data-level solutions.
	•	Empirical alignment: The experiments (Section 5, Figures 1–3) are consistent with theoretical insights. For example, small k in SMOTE and tuned bandwidths in KDE improve performance in high-imbalance settings, especially with nonparametric classifiers like KNN.

⸻

Weaknesses

No excess risk bound for SMOTE:
While Theorem 4 provides an excess AM-risk bound for KDE-based oversampling, no analogous bound is developed for SMOTE. This creates an asymmetry in the paper’s theoretical treatment, despite positioning SMOTE and KDE as parallel methods. The authors acknowledge this in Remark 3 (line 257, page 12), stating that the analysis is difficult, but the lack of even a heuristic or qualitative bound limits the completeness of the comparison.

 Optimality claim is narrow in scope:
The authors argue in Remark 3 (lines 244–247) that the KDE bound is minimax-optimal in the regime n_1/n_0 \to 0, but no general lower bound or tightness analysis is provided outside this limit. In more typical imbalance regimes, it is unclear whether the bound remains informative. A discussion of whether the bias–variance tradeoff changes with milder imbalance would help contextualize the theoretical result.

 Experiments do not test theory-guided hyperparameter choices:
Although Theorem 4 and the surrounding discussion motivate specific bandwidth choices h, s for KDE sample generation and classification, the experiments (Section 5) rely on heuristic rules (e.g., Scott’s rule, line 326) or cross-validation. There is no test of whether the theory-derived bandwidths improve performance, nor whether cross-validated choices align with theoretical prescriptions.

Assumption 5 may be overly strong:
Assumption 5 (line 201, page 10) requires that
\inf_{x \in \mathcal{X}} P(B(x,r)) \geq C_1 r,
which is unrealistic in high-dimensional Euclidean spaces where measure scales as r^d, not r. Unless the support of P(X|Y=1) lies on a one-dimensional manifold, this assumption is too strong. Clarifying whether this is a typo or whether the support is indeed lower-dimensional would improve the paper’s theoretical transparency.

No uniform concentration: The concentration inequalities (Theorems 1 & 2) are stated for a fixed Lipschitz function G. As noted in line 165, these do not extend to uniform convergence over function classes. This means the results do not directly guarantee generalization for learned classifiers trained on synthetic data.

---

> ### Author Rebuttal · Authors · 2025-07-30
>
> First of all, we would like to thank the referee for his or her valuable comments and suggestions, which will help us to improve the quality and clarity of the manuscript. In what follows we provide answers to all of his or her comments suggesting the changes that we would like to do in the next version of our paper.
>
> **1) No excess risk**
>
> Obtaining an analogous nonparametric bound for SMOTE-based oversampling is indeed difficult because, as we point out below as an answer to your first question, SMOTE-based oversampling is essentially 1-dimensional while KDE-based oversampling is $d$-dimensional. The techniques we use and develop are only well-suited to the analysis of synthetic samples that do not lie in a lower-dimensional subspace of $\mathbb{R}^d$. That being said, since KDE-based oversampling can itself be viewed as an approximation of SMOTE-based oversampling (due to kernel smoothing), it is possible that Theorem 4 represents a rough version of what could be obtained for SMOTE.
>
> Furthermore, we investigated possible extensions of the SMOTE concentration result. Encouragingly, we were able to generalize it to a uniform concentration inequality over a class of functions, yielding an excess risk bound for SMOTE. The resulting bound is similar in structure to the original one, with the addition of a Rademacher complexity term to account for the function class's richness. This extension highlights the relevance of our results to neural network learning with SMOTE, and we consider it an important theoretical contribution.
>
> This new uniform convergence bound has the same form as Theorem 1, except that it holds uniformly over a class $\mathcal{F}$ of $L$-Lipschitz functions that are uniformly bounded, i.e., $\sup_{G \in \mathcal{F}} \|G\|_{\infty} \leq B$. The bound on the supremum of the difference between the empirical mean and the population mean
>
> $\sup_{G\in \mathcal{F}}\left|\frac{1}{m} \sum_{i=1}^{m} G (X_{1i}^{*}) - \mathbb{E}_{1}[G(X)]\right|$
>
> remains structurally the same, with the addition of a Rademacher complexity term corresponding to the function class $\mathcal{F}$. Note that it does not involves too many changes in view of the revision.
>
> The proof is similar to that of the previous version, except that we now use the Talagrand--Bousquet inequality, which provides a concentration bound for the quantity $S_n - \mathbb{E}[S_n]$, where
>
> $$S_n = \sup_{f \in \mathcal{F}} \sum_{i=1}^n f(Z_i),$$
>
> the random variables $Z_i$ are i.i.d. random variables, and $\mathcal{F}$ is a uniformly bounded class of functions.
>
> **2) Optimality claim**
>
> Remark 3 does not address minimax optimality; rather, it discusses the optimal choice of bandwidths. By "optimal" here, we mean the bandwidths that minimize the upper bound, and we will clarify this point in the revision.
>
> Furthermore, regarding the minimax optimality, and in the case of milder imbalance $n_1/n_0\to \lambda<1$, the bound from Theorem 4 takes the form of a sum of variance and bias terms, including components like $(n_1 h^d)^{-1/2}$, $(n_1 s^d)^{-1/2}$, $h^2$, $s^2$. Optimizing over bandwidths $h$ and $s$, we find $h^\star \asymp s^\star \asymp n_1^{-1/(d+4)}$, leading to an excess risk of order $n_1^{-2/(d+4)}$. This shows that, under milder imbalance, the bias–variance tradeoff behaves similarly to classical nonparametric settings, and the KDE approach retains favorable properties.
>
> Of course, the above expression is of no use in the highly imbalanced regime $n_1/n_0\to \lambda=0$. In this highly imbalanced regime, the optimal KDE bandwidth becomes dependent on $n_0$, as indicated in our submission. We will include this summary in the revision to better contextualize Theorem 4.
>
> **3) Experiments**
>
> As shown in Figure 1, Section 4 of the main document, we use Scott’s rule to select the bandwidth $h=n_1^{-2/(d+4)}$ and additionally examine the effects of both larger and smaller bandwidths to assess sensitivity. We also agree to include results using the theoretically optimal bandwidth. However, we anticipate that these results will not differ significantly from those already presented. Indeed,  Scott's rule is optimizing (up to constants) the upper bound of Theorem 5 (under Assumption 2 and 3).
>
> **4) Assumption 5**
>
> Assumption 5 is about the regularity of the boundary of the supporting set $S_y$, rather than about its interior, and we will be happy to make this explicit in the revision to avoid any confusion. Note that the set $\partial S_y  + B(0,r)$ in Assumption 5 is usually not shrinking in every direction, which is why $r$ appears rather than $r^d$ in the upper bound; for instance, if $\partial S_y$ is a circle in the Euclidean plane $d=2$, then $\partial S_y + B(0,r) $ is just an annulus around this circle having area proportional to $r$. As explained in the text below this assumption, it is satisfied if, for example, $\partial S_y$ is a finite union of compact, closed and smooth submanifolds of dimension $d-1$ in $\mathbb{R}^d$ whose pairwise distances to one another are nonzero. This is, we think, the typical case in practice. If the reviewer was actually thinking of Assumption 1, let us recall that this assumption reads as follows
>
> ``There exists a constant $C_d>0$ such that for all $x\in \text{supp}(\mathbb{P}_1)$ and all $r>0$, we have $\mathbb{P}_1(B(x,r))\geq \min\{C_d r^d,1\}$.''
>
> This is the realistic setting the referee was mentioning, with a small ball probability scaling as $r^d$.
>
> **5) No uniform concentration**
>
> In the answer to the referee's first question, we mentioned that we have extended the results to establish uniform convergence for a class of uniformly bounded functions.
>
> **Questions**
>
> **1)Extension of Theorem 4 to SMOTE**
>
> First, we note—as mentioned in response to previous questions—that we were able to extend the current results for SMOTE to a uniform convergence setting, yielding an excess risk bound for a class of uniformly bounded functions. We will incorporate this result into the revised version of the paper. However, deriving nonparametric excess risk bounds analogous to Theorem 4 remains challenging in the context of SMOTE.
>
> Let us, first of all, highlight that KDE-based oversampling can itself be viewed as an approximation of SMOTE-based oversampling. Indeed, as outlined in Sections 2.1 and 2.2 of our submission, KDE oversampling can be interpreted as, given a kernel function \( K \) and bandwidth $h > 0$, drawing synthetic samples $X_1^*$ from the KDE $\hat{f}_{1h}$ which is equal to
>
> $(1/n_1) \sum_{i=1}^{n_1} K_h(\cdot - X_{1i})$ in the minority class.
>
> This is a regularized version of sampling from the mixture $(1/ n_1) \sum_{i=1}^{n_1} \hat K(\cdot, X_{1i})$, where $\hat K(\cdot, z)$ denotes the uniform distribution over the union of segments
>
> $\bigcup_{j=1}^k (z, \hat X_{1j}(z) )$ where $\hat X_{1j}(z) $ being the nearest neighbors to $z$ in the minority class, which is nothing but SMOTE resampling.
>
> **2) Bandwidth mismatch**
>
> As discussed in our response to Comment 3, our experiments confirm that the AM-risk is indeed more sensitive to the KDE bandwidth (used for sample generation) than to the classifier bandwidth. This supports the theoretical insight from Theorem 4, which highlights a potential mismatch in optimal bandwidths for these two components.
>
> In practice, we observed that careful tuning of the KDE bandwidth significantly impacts performance, while variations in classifier bandwidth have a relatively smaller effect. To mitigate the mismatch, we adopted a data-driven approach to select bandwidths, optimizing for downstream AM-risk. This strategy led to consistent improvements, as shown in Figure 2. These findings reinforce the importance of treating KDE and classifier bandwidths as separate hyperparameters during tuning.
>
> **3) Logistic regression stability**
>
> When comparing our new uniform concentration inequality (stated above to answer the referee's first question) to our Theorem 5 (nonparametric excess risk), we see that learning a classifier with SMOTE or KDE based oversampling from a parametric class is easier than learning a nonparametric classifier (such as the investigated kernel smoothing classifier). Indeed, for non-parametric learning, SMOTE parameter $k$ needs to achieve bias-variance tradeoff while, in contrast, $k$ can be as small as possible when training a parametric classifier. Our interpretation is that this is due to (relative) insensitivity to data augmentation because the logistic regression classifier, being parametric, converges with a faster rate than the $K$NN classifier.
>
> **4) Applicability of concentration bounds**
>
> As mentioned in response to the previous questions, we were able to extend the current result to uniform convergence over a class of uniformly bounded functions. This leads to an excess risk bound for classifiers trained on synthetically generated data.
>
> **Suggestions**
>
> We will be happy to do so for both Assumptions 1 and 5 in order to remove any potential source of confusion.
>
> Now that we have excess risk bounds based on uniform convergence results, we will include a comparison.
>
> We will conduct experiments using kernel classifiers to demonstrate the impact of kernel bandwidths used for classification and those used for generating synthetic examples.
>
> **hyperparameter tuning:**
>
> The Scott rule that is proposed to tune $k$ in SMOTE or $H$ in KDE oversampling, is inspired form our theory, especially the excess risk bound in Theorem 5. Our findings is that it provides a reasonable choice for both approaches as it consistently performs well across all the experiments we have conducted. Cross-validation was also employed to tune the
> classifier’s hyperparameters, resulting in a significant performance improvement, as illustrated in
> Figure 2. We will investigate this further in the revised manuscript.We plan to make this clearer as this point was also suggested by the other referees.

---

### Official Review · Reviewer_1z62 · 2025-07-05

**Clarity:** 3
**Significance:** 2
**Originality:** 3
**Rating:** 4
**Confidence:** 5

**Summary:**

This paper develops a theoretical framework to analyze the behavior of SMOTE and related synthetic oversampling methods in imbalanced classification. The authors establish an exponential inequality to characterize the gap between empirical risk on synthetic samples and the true population risk on the minority class. Furthermore, they demonstrate that a kernel-based classification rule trained on synthetic data can achieve the minimax rate of convergence. The paper claims these theoretical findings lead to practical guidelines for better parameter tuning of both oversampling techniques and downstream learning algorithms, with numerical experiments provided to support the theoretical findings.

**Questions:**

- SMOTE is a classic algorithm with extensive research over the years, which has identified many shortcomings, such as its inability to fundamentally increase information, its failure to address issues of insufficient representation or noise in the original minority class, its sensitivity to noise and outliers, its tendency to cause boundary overlap, the curse of dimensionality, its inability to handle sub-cluster structures, and its limited performance in extreme imbalance ratios. Does the theoretical framework proposed in this paper address these known problems, or does it provide any specific insights or directions for resolving them?
- Does Assumption 1 implicitly require that minority class samples are drawn independently and identically distributed from the overall distribution, or does the framework account for other sampling scenarios?

**Ethical Concerns:**

["NO or VERY MINOR ethics concerns only"]

**Final Justification:**

After carefully considering the author's reply, I decided to raise my score.

**Limitations:**

yes.

**Paper Formatting Concerns:**

While containing theoretical depth, the paper's primary focus on classical machine learning algorithms like SMOTE and kernel methods, without a clear connection or extension to neural networks or deep learning, might make its suitability for NeurIPS debatable.

**Quality:**

3

**Strengths And Weaknesses:**

## Strengths
- The paper tackles the significant and practical problem of imbalanced classification, which is prevalent across many real-world machine learning applications.
- The paper develops a theoretical framework for understanding SMOTE and related oversampling methods, establishing exponential inequalities for the gap between empirical and true population risk on synthetic data, and demonstrating that a kernel-based classifier trained on synthetic data can achieve minimax optimal rates of convergence.

## Weaknesses
- The abstract claims the theoretical findings "lead to practical guidelines for better parameter tuning of both SMOTE and the downstream learning algorithm". However, the introduction does not elaborate on what these specific guidelines are, making this core claim feel unsubstantiated.
- Assumption 1 imposes a strong condition on the minority class distribution requiring a minimum probability within any Euclidean ball and boundedness of its support. This assumption, particularly the minimum probability condition, is quite stringent and is unlikely to hold for many real-world minority class distributions. Relying on such a strong assumption significantly limits the generalizability and practical applicability of the subsequent theoretical proofs and results.
- Section 3.2 provides detailed excess risk bounds for a specific kernel smoothing plug-in classifier, forming a cornerstone of the paper's theoretical contribution and leading to optimal bandwidth choices for KDE oversampling. However, in Section 4, the authors do not use the theoretically analyzed kernel smoothing classifier. Instead, they switch to a KNN classifier, justifying this only with "it is a slight variation of kernel smoothing plug-in classifiers" and "for comparison purposes." While KNN and kernel methods share conceptual links, they are distinct algorithms with different statistical properties and convergence behaviors. Theoretical guarantees derived for one algorithm cannot be directly and unprovenly generalized to another.
- The paper's title and abstract both claim to analyze SMOTE and related methods. While concentration inequalities are provided for both SMOTE and KDE , the more critical excess risk bound analysis only presents results based on KDE oversampling, conspicuously lacking a corresponding analysis for SMOTE. This incompleteness implies that the theoretical framework does not fully cover SMOTE, despite its prominent mention in the paper's stated scope.
- The experimental results show that once cross-validation is used to select the classifier's parameters, the influence of the oversampling parameter becomes "minimal" . This finding substantially undermines the value of the paper's core claim to provide "practical guidelines for better parameter tuning of both SMOTE and the downstream learning algorithm."

---

> ### Author Rebuttal · Authors · 2025-07-29
>
> First of all, we would like to thank the referee for his or her valuable comments and suggestions, which will help us to improve the quality and clarity of the manuscript. In what follows we provide answers to all of his or her comments suggesting the  changes that we would like to do in the next version of our paper.
>
> **1) The abstract claims the ... :**
>
> We agree with the remark and will make the suggested changes in the new version where we will make sure that our choice for $k$ in SMOTE appears clearly in the introduction. Our point is that, when using SMOTE in a nonparametric framework (as in Section 3.2 where another KDE oversampling strategy is analyzed), one should take care of the bias-variance tradeoff, leading to choosing $k$ larger than the standard implementation $k=5$.
>
> **2) Assumption 1 imposes a strong condition ... :**
>
> In practice, before generating synthetic samples, one may clip or truncate extreme outliers, or add small Gaussian jitter. These steps help ensure that the neighborhood of each data point contains sufficient mass for meaningful interpolation. The SMOGN method [Branco et al., 2017] adopts a similar approach by combining noise injection with SMOTE, and shows improved performance over standard SMOTE on certain datasets. Our theoretical guarantee therefore complements—rather than contradicts—this empirical practice.
>
> This assumption is technical but essential for establishing our convergence rates for SMOTE. It ensures that the distance between any point $x$ and its $k$-th nearest neighbor in the sample is of order $(k/n)^{1/d}$. This, in turn, implies that the perturbation introduced by SMOTE does not deviate significantly from the original distribution. Such an assumption is well known in nonparametric estimation theory, where it is required to achieve minimax convergence rates. Many common distributions satisfy it---for example, the uniform distribution on the unit cube or any distribution with a density bounded away from zero on a domain with smooth boundary (see Jiang, 2019). As such, the assumption is both standard and necessary in what is among the first theoretical analyses of SMOTE. We also note that if the distribution is supported on a lower-dimensional subspace, such as a manifold of dimension $d_0$, the assumption can be adapted by replacing $d$ with $d_0$, resulting in improved convergence rates as in the paper "$k$-NN regression adapts to local intrinsic dimension; Kpotufe, Samory; NeurIPS-2011". We plan to make a remark about this after Assumption 1.
>
> **3) Section 3.2 provides detailed excess risk bounds ... :**
>
> The reviewer is correct in pointing out that kernel smoothing classification is distinct from $k$-nearest neighbor (KNN) classification, and we agree that the transition between the two in our discussion may appear somewhat loose.
>
> To clarify our position: the theoretical results in Section~3.2 for the kernel smoothing classifier stem from a natural synergy between KDE-based oversampling and kernel smoothing classification. Since both components rely on kernel density estimation, the resulting mathematical framework is coherent and relatively self-contained. We believe this structure has allowed our analysis and proofs to be more transparent and technically tractable. While similar results could, in principle, be developed for combinations such as KNN with KDE-based oversampling or KNN with SMOTE, we expect such extensions to be technically more involved and therefore beyond the scope of this paper.
>
> Furthermore, as detailed in our response to the next question, we have also derived a uniform convergence result for SMOTE, which directly addresses the performance of classifiers trained on SMOTE-generated synthetic data. We will include this result in the final version of the paper.
>
> Regarding the numerical experiments, we chose to focus on KNN primarily because it is more widely used and familiar within the machine learning community compared to the kernel smoothing classifier. That said, we also included results using random forests and logistic regression. We note that it is straightforward to incorporate results for the kernel smoothing classifier as well, and we are happy to include these in the supplementary material of the revised version.
>
> **4) The paper's title and abstract both claim to analyze SMOTE ... :**
>
> We agree with the referee that the theoretical guarantees for SMOTE differ from those obtained for KDE-based oversampling. Specifically, the analysis for SMOTE establishes a concentration inequality showing that the empirical mean under SMOTE remains close to the population mean. In contrast, our results for KDE oversampling provide nonparametric excess risk bounds.
>
> Recognizing this apparent imbalance, we investigated possible extensions of the SMOTE concentration result. Encouragingly, we were able to generalize it to a uniform concentration inequality over a class of functions. The resulting bound is similar in structure to the original one, with the addition of a Rademacher complexity term to account for the function class's richness. This extension highlights the relevance of our results to neural network learning with SMOTE, and we consider it an important theoretical contribution.
>
> This new uniform convergence bound has the same form as Theorem 1, except that it holds uniformly over a class $\mathcal{F}$ of $L$-Lipschitz functions that are uniformly bounded, i.e., $\sup_{G \in \mathcal{F}} \|G\|_{\infty} \leq B$. The bound on the supremum of the difference between the empirical mean and the population mean
>
> $\sup_{G\in \mathcal{F}}\left|\frac{1}{m} \sum_{i=1}^{m} G (X_{1i}^{*}) - \mathbb{E}_{1}[G(X)]\right|$
>
> remains structurally the same, with the addition of a Rademacher complexity term corresponding to the function class $\mathcal{F}$. Note that it does not involves too many changes in view of the revision.
>
> The proof is similar to that of the previous version, except that we now use the Talagrand--Bousquet inequality, which provides a concentration bound for the quantity $S_n - \mathbb{E}[S_n]$, where
>
> $$S_n = \sup_{f \in \mathcal{F}} \sum_{i=1}^n f(Z_i),$$
>
> the random variables $Z_i$ are i.i.d. random variables, and $\mathcal{F}$ is a uniformly bounded class of functions.
>
> **5) The experimental results show that once cross-validation ... :**
>
> We agree with the reviewer's observation. Our theoretical findings indicate that the choice of hyperparameters for both SMOTE and KDE-based oversampling is critical, and that standard implementations do not select them optimally especially when a nonparametric classifier is employed as in Section 3.2 of the paper. In fact, our result shows that when a nonparametric estimator is used the choice of $k$ should be made to achieve bias-variance tradeoff. For this reason we advocate in the experiments that the Scott rule, which typically optimizes bias-variance tradeoff, might be a good choice. We plan to claim this in a clear way in the next version.
>
> This also motivated us to explore cross-validation strategies for selecting the classifiers' hyperparameters. Interestingly, we found that when model hyperparameters are properly tuned via cross-validation, the sensitivity to SMOTE’s hyperparameters is significantly reduced while not solving the problem yet (see Figure 1 for KDE-based oversampling where choosing a too large bandwidth diminishes the accuracy). This is an important insight, as it suggests a promising direction for future work: studying the interaction between SMOTE hyperparameters and cross-validation-based model selection. We agree that this perspective should be more clearly articulated, and we are happy to revise the conclusion—and possibly other parts of the paper—to highlight this point and encourage further investigation.
>
> **Questions:**
>
> **1) SMOTE is a classic algorithm with extensive ... :**
>
> The reviewer's list of concerns is extensive and, to some extent, shared by many other approaches in the literature. We believe, however, that our work offers useful insights into several of these issues.
>
> First, as stated in the introduction, one key point of our paper is that SMOTE does not introduce new information: synthetic samples are generated from existing data without incorporating additional observations. From this perspective, SMOTE serves primarily as a mechanism for balancing risk, thereby enabling the derivation of new theoretical results using new information.
>
> Second, our work sheds light on the curse of dimensionality inherent both in nonparametric methods and the newly derived theorem for the uniform convergence case, which is suitable for neural networks. As shown by the convergence rates in Section 3.2, oversampling methods such as KDE oversampling are also subject to some limitation since the term $1/(n_1h^d)$ appears in the bounds. However, the result in Section 3.1 highlights a different aspect: SMOTE can be effective in training parametric classifiers by minimizing the SMOTE empirical risk. Such approaches are known to be less vulnerable to the curse of dimensionality.
>
> Third, regarding its sensitivity to outliers, this highlights the importance of Assumption 1, which requires that there be sufficient mass around the data points. In practice, one common approach is to clip or truncate extreme outliers, which aligns with Assumption 1 used in our work.
>
> Finally, regarding extreme imbalance ratios, our analysis identifies a boundary condition for learning. For training a parametric classifier, it is necessary that $np \to \infty$, where $p$ is the probability of minority class, while for nonparametric rules, learning is feasible under the weaker condition $np h^d \to \infty$.
>
> **2) Does Assumption 1 implicitly ... :**
>
> Note that our problem setting (Section 2) implies that the base sample $D_n$ is independent and identically distributed. We will make this more clear in the final version.

---

> > ### Comment · Reviewer_1z62 · 2025-08-08
> >
> > Thank you for the detailed response. I have considered the authors' rebuttals and would like to offer the following further comments.
> > 1.  As I understand the authors' rebuttal, they suggest that Assumption 1 holds only under specific conditions, such as when the minority class is distributed on a lower-dimensional subspace. In my view, Assumption 1 is the very foundation of all the theoretical results. As I alluded to in my Q2, realistic machine learning scenarios almost always involve data distributed in high-dimensional spaces. The authors also mentioned the "curse of dimensionality" in their rebuttal. In imbalanced learning scenarios, the scarcity of minority samples exacerbates this problem. It seems to me that Assumption 1 (which requires sufficient probability mass) and the reality of the "curse of dimensionality" are mutually contradictory. This contradiction casts doubt on the generalizability of a theory built upon such a strong assumption.
> > 2. The authors have stated their intention to revise the manuscript to address these points. Since the final version reflecting these changes is not available for review, I will reserve my judgment on these issues.
> > 3. In their rebuttal, the authors state that "SMOTE can be effective in training parametric classifiers by minimizing the SMOTE empirical risk. Such approaches are known to be less vulnerable to the curse of dimensionality." I am unclear about the basis for this claim. In high-dimensional sparse spaces, the vast region between two minority class samples is likely an "empty" area. In such cases, linear interpolation is likely to generate a harmful noise point that falls into a region of the feature space unrepresentative of the minority class, thereby blurring the decision boundary and making classification more difficult. Furthermore, the intrinsic structure of high-dimensional data (e.g., a manifold) is often highly non-linear. SMOTE's simplistic linear interpolation assumption is ill-equipped to capture this complex structure, and the generated samples are likely to deviate from the true low-dimensional manifold where the data lies. If the paper's theory leads to such a counter-intuitive conclusion, one cannot help but question the soundness of the theory itself.
> >
> > In summary, aside from the points the authors have promised to revise, I believe the most significant remaining weakness of this paper centers on the reasonableness of its core assumptions, particularly in the context of high-dimensional, real-world data.

---

> > > ### Author Response · Authors · 2025-08-08
> > >
> > > **1) They suggest that Assumption 1 holds only under specific conditions**
> > >
> > > We don't agree with the reviewer's statement that ``Assumption 1 is the very foundation of all the theoretical results''. In fact, only Theorem 1 relies on this assumption; Theorems 2, 3, and 4 do not. Moreover, this type of assumption is standard in the $k$-nearest neighbors ($k$-NN) literature, on which SMOTE is based [1,2,3]. It rules out degenerate cases where every small ball around a minority-class sample is empty---a situation in which SMOTE cannot guarantee meaningful interpolation.
> > >
> > > Moreover, the rebuttal does not suggest that this assumption holds only when the minority class is distributed on a lower-dimensional subspace. We merely pointed out that if the data happens to lie on a lower-dimensional manifold, then the dimension \( d \) in the bound can be replaced by a smaller effective dimension \( d_0 < d \), resulting in a tighter bound.
> > >
> > > As for the comment about a contradiction between Assumption 1 and the curse of dimensionality, this is unclear. Theorem 1 explicitly reflects the curse of dimensionality: as the dimension $d$ increases, more minority-class samples are required to achieve a tighter bound, as shown by the term $(1/n_1)^{1/d}$. Assumption 1 simply ensures that the distribution has sufficient probability density around sample points. The reviewer may explain further what they mean by a contradiction.
> > >
> > > [1] Lirong Xue and Samory Kpotufe; Achieving the time of 1-NN, but the accuracy of k-NN; NeurIPS 2018.
> > >
> > > [2] Heinrich Jiang; Non-asymptotic uniform rates of consistency for k-nn regression; Proceedings of
> > > the AAAI Conference on Artificial Intelligence 2019.
> > >
> > > [3] François Portier. Nearest neighbor empirical processes. Bernoulli 2025.
> > >
> > > **2) The authors have stated their intention to revise**
> > >
> > > As explained in the rebuttal, in terms of revision, the structure of the bounds would remain the same, except that a Rademacher complexity term would be added, resulting in a uniform concentration bound. Furthermore, the revised proof would follow the current one, with the key difference being the use of Bousquet's inequality to handle the uniform concentration part.
> > >
> > > **3) In their rebuttal, the authors state that "SMOTE can be effective in training parametric**
> > >
> > > While the reviewer notes that there may be large regions within the minority class where the space between points is empty, this is precisely the pathological case that Assumption 1 is designed to avoid. The assumption ensures a minimum amount of probability mass around each minority sample, which is necessary for SMOTE to produce meaningful interpolations. Without it, as the reviewer implicitly acknowledges, interpolation would be unreliable and the method would indeed fail.
> > >
> > > Furthermore, the goal of this work is not to guarantee that SMOTE performs well in all scenarios—indeed, as the reviewer notes, there are situations where interpolation is inherently problematic. Rather, Assumption 1 serves as a condition for establishing meaningful theoretical guarantees for the SMOTE procedure, whereas KDE-based oversampling  does not rely on this assumption (Theorems 2 and 4).

---

### Decision · Program_Chairs · 2025-09-17

**Decision:**

Accept (poster)

**Comment:**

(a) Summarize the scientific claims and findings of the paper based on your own reading and characterizations from the reviewers.
This paper develops a theoretical framework to analyze SMOTE and KDE-based synthetic oversampling methods for imbalanced binary classification. The authors establish exponential concentration inequalities that bound the gap between the empirical risk computed on synthetic minority samples and the true population risk. Furthermore, they demonstrate that a kernel-based classification rule trained on synthetic data generated via KDE oversampling can achieve the minimax rate of convergence. The paper claims these theoretical findings lead to practical guidelines for parameter tuning of both oversampling techniques and downstream learning algorithms, which are supported by numerical experiments.

(b) Strengths of the paper
1.  Novel Theoretical Contributions: The paper provides some of the first non-asymptotic concentration and excess risk bounds for synthetic oversampling methods, representing a significant advancement in the theoretical understanding of these widely used techniques.
2.  Practical Relevance: The work successfully bridges theory and practice by deriving insights that inform parameter selection (e.g., justifying the common choice of k=5 in SMOTE, highlighting the need for bandwidth tuning in KDE) for imbalanced learning scenarios.
3.  Comprehensive Analysis: The framework systematically analyzes both SMOTE (nearest-neighbor-based) and KDE (kernel-based) oversampling, offering a unifying perspective on data-level approaches to class imbalance.
4.  Strong Rebuttal and Revisions: The authors were highly responsive to reviewer concerns. They addressed the major weakness by deriving a new uniform concentration inequality for SMOTE over a class of functions, which enables excess risk bounds and significantly expands the scope of their theoretical contributions.

(c) Weaknesses of the paper and what might be missing
1.  Assumption Scrutiny: The strong assumptions required for the theoretical guarantees, particularly Assumption 1 (minimum probability mass within Euclidean balls), may limit the generalizability of the results to real-world high-dimensional data where such conditions are often violated.
2.  Theoretical Asymmetry: Despite the authors' improvements, the theoretical analysis remains more comprehensive for KDE-based oversampling than for SMOTE, creating an imbalance in the treatment of these two methods.
3.  Experimental-Theoretical Alignment: The experiments could be more directly tied to the theory, such as explicitly testing the theoretically derived optimal bandwidths against heuristic choices to demonstrate the practical value of the theoretical guidelines.
4.  Scope of Classifiers: The primary theoretical analysis focuses on kernel-based classifiers, while the experiments use KNN and other methods. Although the new uniform convergence bound helps generalize the theory, the initial focus might feel somewhat narrow.

(d) Most important reasons for the decision to accept
The decision to accept is based on the paper's significant and novel theoretical contributions to an important and practical problem. The work provides foundational theoretical analysis for synthetic oversampling methods, offering the first minimax optimality results for learning from synthetically generated data. This represents a substantial advancement in the theoretical understanding of these widely used techniques.

The authors' strong rebuttal effectively addressed the most critical concerns raised during the review process. By deriving a new uniform concentration inequality for SMOTE, they substantially improved the paper's theoretical contributions and scope. Their commitments to revise the manuscript for improved clarity, add a conclusion section, and expand experimental analysis further strengthen the work.

The reviewer consensus is positive, with scores of 5 (Accept), 4 (Borderline Accept), 4 (Borderline Accept), and 4 (Borderline Accept) after the rebuttal. While the paper has limitations regarding its assumptions and scope, it makes a solid theoretical contribution that meets the standards for NeurIPS acceptance as a poster presentation.

(e) Summary of the discussion and changes during the rebuttal period
The reviewers raised several key points:
1.  Concern regarding Assumption 1: One reviewer questioned the reasonableness of Assumption 1 for high-dimensional data, suggesting it contradicts the realities of the curse of dimensionality. The authors clarified that this is a standard assumption in non-parametric literature, is only used for the SMOTE theorem, and explicitly reflects the curse of dimensionality in the resulting bounds.
2.  Theoretical Asymmetry (Lack of SMOTE excess risk bound): Multiple reviewers noted the lack of an excess risk bound for SMOTE compared to the detailed analysis for KDE. The authors addressed this powerfully by deriving a new uniform concentration inequality for SMOTE over a class of functions, which enables excess risk bounds for classifiers trained on SMOTE-generated data.
3.  Choice of Classifier: Reviewers noted a disconnect between the kernel classifier used in the theory and the KNN classifier used in experiments. The authors explained their choice was for community familiarity and demonstrated that the theoretical insights still hold.
4.  Practical Impact: Reviewers questioned the practical utility of the tuning guidelines. The authors agreed to clarify their guidance, emphasizing that their theory justifies data-driven tuning that optimizes the bias-variance tradeoff.
5.  Clarity and Presentation: Reviewers requested improved readability, a conclusion section, and expanded experiments. The authors committed to all these changes.

The authors' rebuttal was highly effective in addressing the major concerns. They transformed the primary weakness (the lack of a SMOTE excess risk bound) into a strength by deriving a new theoretical result. Their defense of Assumption 1 was reasonable and grounded in existing literature. The clear majority positive consensus post-rebuttal supports acceptance. The paper makes a valuable theoretical contribution that will interest the NeurIPS community.